# The Paradoxical Effect Hypothesis of Abused Drugs in a Rat Model of Chronic Morphine Administration

**DOI:** 10.3390/jcm10153197

**Published:** 2021-07-21

**Authors:** Yinghao Yu, Alan Bohan He, Michelle Liou, Chenyin Ou, Anna Kozłowska, Pingwen Chen, Andrew Chihwei Huang

**Affiliations:** 1Department of Psychology, Fo Guang University, Yilan County 26247, Taiwan; yuyinghao3000@yahoo.com.tw (Y.Y.); 1012803@mail.fgu.edu.tw (A.B.H.); chenyinou1990@gmail.com (C.O.); seasqseasq@gmail.com (P.C.); 2Department of Biotechnology and Animal Science, National ILan University, Yilan 26047, Taiwan; 3Institute of Statistical Science, Academia Sinica, Taipei 11529, Taiwan; michelleliou@gmail.com; 4Department of Human Physiology, School Medicine, Collegium Medicum, University of Warmia and Mazury in Olsztyn, Warszawska Av 30, 10-082 Olsztyn, Poland; kozlowska.anna@uwm.edu.pl

**Keywords:** morphine, the paradoxical effect hypothesis of abused drugs, reward, aversion, dual effect

## Abstract

A growing body of studies has recently shown that abused drugs could simultaneously induce the paradoxical effect in reward and aversion to influence drug addiction. However, whether morphine induces reward and aversion, and which neural substrates are involved in morphine’s reward and aversion remains unclear. The present study first examined which doses of morphine can simultaneously produce reward in conditioned place preference (CPP) and aversion in conditioned taste aversion (CTA) in rats. Furthermore, the aversive dose of morphine was determined. Moreover, using the aversive dose of 10 mg/kg morphine tested plasma corticosterone (CORT) levels and examined which neural substrates were involved in the aversive morphine-induced CTA on conditioning, extinction, and reinstatement. Further, we analyzed c-Fos and p-ERK expression to demonstrate the paradoxical effect—reward and aversion and nonhomeostasis or disturbance by morphine-induced CTA. The results showed that a dose of more than 20 mg/kg morphine simultaneously induced reward in CPP and aversion in CTA. A dose of 10 mg/kg morphine only induced the aversive CTA, and it produced higher plasma CORT levels in conditioning and reacquisition but not extinction. High plasma CORT secretions by 10 mg/kg morphine-induced CTA most likely resulted from stress-related aversion but were not a rewarding property of morphine. For assessments of c-Fos and p-ERK expression, the cingulate cortex 1 (Cg1), prelimbic cortex (PrL), infralimbic cortex (IL), basolateral amygdala (BLA), nucleus accumbens (NAc), and dentate gyrus (DG) were involved in the morphine-induced CTA, and resulted from the aversive effect of morphine on conditioning and reinstatement. The c-Fos data showed fewer neural substrates (e.g., PrL, IL, and LH) on extinction to be hyperactive. In the context of previous drug addiction data, the evidence suggests that morphine injections may induce hyperactivity in many neural substrates, which mediate reward and/or aversion due to disturbance and nonhomeostasis in the brain. The results support the paradoxical effect hypothesis of abused drugs. Insight from the findings could be used in the clinical treatment of drug addiction.

## 1. Introduction

Abused drugs have multiple effects—reward and aversion—that are balanced with each other, and thereby it causes drug addiction and dependence [1]. In the past few years, a growing body of studies has reported that abused drugs possess reinforcement or reward to drive abusers that continuously take drugs [2,3]. These studies used the tasks of drug self-administration [4] and conditioned place preference (CPP) [5] to induce craving and relapse behaviors. However, little research demonstrated the aversive effect induced by abused drugs, and this line of research often uses conditioned taste aversion (CTA) [6] or conditioned place aversion (CPA) [7] tasks to imitate avoiding drug-taking behaviors. In contrast to the reward, the abused drug’s aversive property is also crucial for drug dependence.

Concerning the recent issue of the CTA, Grigson and her colleagues examined how the suppression of preferred taste (conditioned stimulus; CS) was caused by the reward value of abused drugs (unconditioned stimulus; US) outweighing that of the preferred taste, and the CTA is due to the rewarding property of abused drugs, which is called the reward comparison hypothesis [8]. Our further evidence challenged the reward comparison hypothesis and demonstrated that abused drugs induced CTA resulting from the aversion of abused drugs but not its reward [9,10,11]. Moreover, we suggest that abused drugs, including amphetamine [10] and ethanol [11], simultaneously induced reward and aversion, called the paradoxical effect hypothesis of abused drugs [12]. However, no research examines whether morphine can simultaneously induce reward and aversion as well as amphetamine [10] and ethanol [11]. Therefore, Experiment 1 addressed this issue and tested various doses of morphine for aversion in a CTA task and reward in a CPP task.

Previous studies of abused drugs have shown that certain brain areas mediated reward or aversion or even both effects. For example, morphine-induced CPP could increase dendritic spine density in the ventral hippocampus (i.e., the CA1 and dentate gyrus (DG)) [13]. Repeated treatments of amphetamine produced stereotypical behavior and increased mRNA BDNF levels in the basolateral amygdala (BLA) and the rostral piriform cortex (rPC) [14]. Morphine, studied in the present work, binds to the mu-opioid receptor expressed on opiate neurons and interacts with the orexin receptor belonging to the G protein-coupled receptor, activating orexin signaling [15]; a recent study showed that orexin signaling within the lateral hypothalamus (LH) mediated synaptic plasticity in the reward effect of abused drugs [16]. The medial prefrontal cortex (the prelimbic(PrL) cortex and infralimbic (IL) cortex) regulated the inhibition of drug-seeking behavior through the dopamine neurons in the IL [17] or the GABA neurons in the PrL [18]. The anterior cingulate cortex governed the reward signals from the mesolimbic dopamine system [19], and dysfunctions of the anterior cingulate cortex produced impulsive and compulsive behaviors in drug addiction [20]. Fewer neural substrates contribute to the aversive effect of drug addiction. With respect to the central amygdala (CeA), corticotrophin-releasing factor secretions are involved in anxiety or aversion inwithdrawal symptoms [21]. Besides, some neural substrates are involved in the reward and aversion of abused drugs. For example, the D1 receptor (within the BLA projection from the VTA dopamine neurons) mediates the enhancement function of opiate-related reward processes, and the D2 receptor (also within the BLA) governs the aversive effects of opiate dependence and withdrawal [22,23]. Injections of D1 antagonist SCH23390 into the nucleus accumbens (NAc) disrupted abused drug-induced rewarding CPP [24] and aversive-conditioned taste suppression [25]. The DG of the hippocampus regulated the aversive effect of opiate withdrawal-associated memory [26] and the rewarding effect of CPP [13]. Furthermore, the previous studies showed that the subareas Cg1 [19], PrL [18], and IL [27] of the mPFC; CA1, CA2, and CA3 of the hippocampus [28], the LH [29], and the PC [30] contributed to the reinforcement process or the rewarding effect of the abused drugs. Moreover, the DG of the hippocampus [26], the NAc [31], and the BLA [32] are involved in the reward and aversion of abused drugs. Growing evidence shows that the CeA only mediates the aversion of abused drugs for drug addiction [33,34] (Figure 1). In Experiment 2, the aversive dose of morphine tested whether the selected neural substrates (such as the mPFC (e.g., Cg1, PrL, and IL), the hippocampus (e.g., CA1, CA2, CA3, and DG), the NAc, the LH, the amygdala (e.g., BLA and CeA), and the PC) mediate morphine-induced conditioned taste suppression in conditioning, extinction, and reinstatement using c-Fos and p-ERKimmunohistochemical staining.

Some review papers have reported that drug-taking was affected by the balance between reward and aversion [1,35,36]. However, these studies did not examine the brain mechanisms. We compared the previous data of rewarding neural substrates in drug addiction to ours [37] and demonstrated the paradoxical effect or nonhomeostasis and disturbance under morphine-induced conditioned taste suppression. Levels of corticosterone secreted from the adrenal gland were related to stress and aversion [38]. This study also tested plasma corticosterone levels to clarify morphine-induced conditioned taste suppression due to aversion but not reward.

Altogether, the study concerned: (a) which doses of morphine can simultaneously produce reward in CPP and aversion in CTA tasks in Experiment 1. (b) Experiment 2 used the aversive dose of morphine to examine which neural substrates were involved in the aversive morphine-induced conditioned suppression and testing of corticosterone levels in conditioning, extinction, and reinstatement. (c) Analyzing c-Fos and p-ERK expression demonstrated the paradoxical effect—reward and aversion and nonhomeostasis or disturbance by morphine-induced conditioned taste suppression.

## 2. Material and Methods

### 2.1. Behavioral Procedure

The study had two separate experiments, and flowcharts areshown in Appendix A. Experiment 1 was to test various doses of morphine to determine which dose of morphine caused reward, aversion, or both effects. Furthermore, Experiment 2 was conducted using a morphine dose which only induced aversion but not reward, to induce conditioned suppression in behavior, and it examined which neural substrates were involved in the morphine-induced conditioned suppression in conditioning, extinction, and reinstatement phases.

#### 2.1.1. Experiment 1: Testing Different Doses of Morphine for Reward and Aversion

In Experiment 1, all rats were raised in the home cage to habituate to the colony room for 7 days in the adaptation phase. Later, rats were given a water deprivation regimen for 5 days (Days 8–12). During this period, the rats were deprived of water for 23.5 h/day. Then, they were trained to drink water in the lickometer device for 15 min and drink water for 30 min in the home cage in the evening. On the last day (Day 12), all rats were allowed to explore all CPP compartments for 10 min. During the conditioning phase (Days 13–22), the rats were assigned to 0 mg/kg (*n* = 17), 10 mg/kg (*n* = 12), 20 mg/kg (*n* = 13), 30 mg/kg (*n* = 14), and 40 mg/kg (*n* = 14) morphine groups on five drug-paired and five drug-unpaired trials. Note that some of the rats (0 mg/kg (*n* = 11), 10 mg/kg (*n* = 9), 20 mg/kg (*n* = 9), 30 mg/kg (*n* = 8), and 40 mg/kg (*n* = 12)) would be further tested in the CPP tasks. In the conditioning phase, the rats were subjected to morphine or saline-paired treatments on even days. In each of the five drug-paired sessions (odd days), the rats were given 0.1% saccharin solution for 15 min and intraperitoneally injected with morphine or normal saline. In each of the five unpaired sessions (even days), all groups were injected with saline immediately prior to being placed in the other compartment in which they had not been previously confined and were left in the compartment for 30 min. Exposure to the drug-paired and unpaired compartments in the morphine and saline groups was conducted in a counterbalanced way. For CTA training, the consumption of a 0.1% saccharin solution was measured for 15 min on even days. CTA training was not performed on the odd days of the conditioning phase. The CTA training was conducted for a total of five trials. For CPP testing, a rat was put into the CPP apparatus for 15 min without the wood partitions on the testing phase (Day 23). The amount of time each rat spent in each compartment was measured by stopwatches. When any three of a rat’s four limbs were placed in a specific compartment, the time would be measured for that compartment (Appendix A).

#### 2.1.2. Experiment 2: Which Neural Substrates Were Involved in the Reward, Aversion, or Even Both Effects Induced CTA by Morphine

Based on the result of Experiment 1, we found that 10 mg/kg of morphine was the appropriate dose to produce aversion in CTA but not reward in CPP. Experiment 2 used 132 rats. Note that, 48 rats were tested in the behavioral tasks in conditioning (*n* = 16), extinction (*n* = 16), and reinstatement (*n* = 16) phases. Seventy-two rats were subjected to immunohistochemical staining to label c-Fos (*n* = 36) and p-ERK (*n* = 36) expression in conditioning, extinction, and reinstatement, and each of conditioning, extinction, and reinstatement phases used 12 rats for saline and morphine groups (*n* = 6 per group), respectively. Twelve rats were subjected to the Enzyme-Linked Immunosorbent Assay (ELISA) corticosterone assessment for the saline (*n* = 6) and morphine groups (*n* = 6) in baseline, conditioning, extinction, and reinstatement phases. The present study manipulated 10 mg/kg morphine to pair with a 0.1% saccharin solution intake and then produced conditioned suppression for the saccharin solution. In the adaptation phase, all rats were raised in their home cages in the colony room for 7 days to habituate them. Later, the rats received a waterdeprivation regimen for 2 days, which involved 23.5 h/day of water deprivation. The rats were trained to drink water using the lickometer device for 15 min and drink water for 30 min in the home cage in the evening. In the conditioning phase, all rats were free to drink a 0.1% saccharin solution for 15 min and were then given an injection of saline or 10 mg/kg morphine in each of the 5 daily trials. On the last day, the rats were sacrificed, and their brains were removed. The c-Fos and p-ERK expressions of specific brain areas could be labeled using the immunohistochemical staining method. In the extinction phase, other rats experienced the same conditioning procedure for each of the five trials. After conditioning, the rats were allowed to drink a 0.1% saccharin solution for each trial for 15 min without any drug injections. The rats’ brains were then removed so that the c-Fos and p-ERK expressions could be labeled through the immunohistochemical staining method. In the reinstatement phase, the rats underwent adaptation for 7 days, water deprivation for 3 days, waterdrinking training (using the lickometer) for 2 days, CTA conditioning for 5 days, and extinction for 5 days. On the last day of the extinction phase, the saline and morphine groups were given injections of saline or 10 mg/kg morphine, respectively, as part of the reinstatement procedure. On the next day, all rats were allowed to drink the 0.1% saccharin solution. After that, the rats were sacrificed, and their brains were removed. The c-Fos and p-ERK expressions could be labeled using the immunohistochemical staining method. Furthermore, trunk blood was drawn from each rat during the last days of the adaptation (baseline), conditioning, extinction, and reinstatement phases to measure the corticosterone levels in the plasma (Appendix A).

### 2.2. ELISA: Corticosterone Assessments

The blood samples were collected in a heparinized tube (75 USP Units) and stored temporally on ice. These samples were centrifuged for 15 min at 3000× rpm and 4 °C. These blood samples were stored at −20 °C until they were assayed. The plasma corticosterone levels were determined in duplicate using a commercial immunoassay kit (Corticosterone EIA kit; Cayman Chemical Company, Ann Arbor, MI, USA) with 96-well microtiter plates. The corticosterone levels were assessed with a photometric microplate reader (Metertech Company, Taipei, Taiwan) at 405 nm. This kit sensitivity was 40 pg/mL, which is lower than the lowest detectible value. The coefficients of the duplicate variation within and between assays were <6.0%.

### 2.3. Immunohistochemical Staining: c-Fos and p-ERK

The rats were injected with a sodium pentobarbital overdose. Later, 100 mL of 0.1 M sodium phosphate-buffered saline (PBS) solution was injected. Then, 400 mL of 4% paraformaldehyde in a 0.1 M PBS buffer was supplied for perfusion. For cryoprotection, the rat’s brain tissue was dissected, post-fixed, and delivered to 30% sucrose solution until the sample sank to the bottom of the solution. Section of 40 μm were sliced through the whole brain [39]. All slices were performed for p-ERK or c-Fos immunoreactivity. The floating brain slices were washed for 15 min once in 0.1 M PBS, permeabilized in 3% H2O2 for 1 h, and washed for 20 min in 2% PBS tween-20 (PBST) three times. Finally, the brain slice was soaked for 1 h in 3% normal goat serum and 1% bovine serum albumin. After washing PBST for 15 min twice, the slices were incubated at 4 °C overnight to perform p-ERK labeling using rabbit anti-p-ERK (Genetex Inc., Irvine, CA, USA, GTX50274, 1:500) and c-Fos labeling with rabbit anti-Fos antibody (Santa Cruz Biotechnology Inc., Dallas, TX, USA, SC-52, 1:1000). Then, the brain slices were incubated with a biotinylated goat anti-rabbit secondary antibody (Vector Lab BA-1000, 1:500) for 1 h. After washing the slices with PBS solution for 10 min, the secondary antibody was conducted with the ABC kit (Vector Lab ABC Kit, PK-6100). The ImageJ program counted the positive expression of p-ERK or c-Fos visually at 20 magnifications for each slice for the whole brain [40]. The number of c-Fos or p-ERK protein expressions was averaged in a selected brain area for each group.

### 2.4. Drugs

Sodium saccharin and sodium chloride were bought from the Sigma-Aldrich Company in the USA (St. Louis, MO, USA). Sodium saccharin was dissolved in distilled water and prepared in a 0.1% saccharin solution. Morphine hydrochloride was bought from the Food and Drug Administration, Ministry of Health and Welfare, Executive Yuan in Taipei in Taiwan. The control group was given 0 mg/kg morphine, and this group was intraperitoneally injected with normal saline at 1 mL/kg. Morphine was dissolved in normal saline at a concentration of 10 mg/mL. Morphine was intraperitoneally injected at a 1, 2, 3, or 4 mL/kg volume, and then they were transferred and served atthe doses of 10, 20, 30, and 40 mg/kg morphine. Although the injection volumes of morphine were different, the doses were injected at 10, 20, 30, and 40 mg/kg into the body, respectively.

## 3. Results

### 3.1. Experiment 1: Different Doses of Morphine Testing for Reward and Aversion

A 5 × 5 mixed two-way ANOVA indicated that significant effects occurred for the group (F4, 65 = 8.62, *p* < 0.05), session (F4, 260 = 24.95, *p* < 0.05), and interaction of group and session (F16, 260 = 4.99, *p* < 0.05). A post hoc Tukey test indicated that the mean intake volume of the 10 mg/kg, 20 mg/kg, 30 mg/kg, and 40 mg/kg morphine groups had significantly decreased more than that of the saline group from Sessions 2–5 (*p* < 0.05). Therefore, doses equal to and higher than 10 mg/kg of morphine could produce the conditioned suppression of saccharin solution intake (Figure 2).

A dependent *t*-test indicated that the spent time was significantly increased for the drug-paired side in the 20 mg/kg, 30 mg/kg, and 40 mg/kg morphine groups (*p* < 0.05; Figure 3B). However, there were no significant differences between the 0 mg/kg and 10 mg/kg morphine groups between the drug-paired and drug-unpaired sides (*p* > 0.05; Figure 3B). No group showed significant differences between the drug-paired and drug-unpaired sides (*p* > 0.05; Figure 3A).

Therefore, morphine simultaneously induced reward in CPP and aversion in CTA when the dose was higher than 20 mg/kg. The dose of 10 mg/kg of morphine was the most appropriate for aversive CTA learning (Table 1), and this dose of 10 mg/kg morphine was used in Experiment 2.

### 3.2. Experiment 2: Assessments of Conditioning, Extinction, and Reinstatement Behavior

A 2 × 5 mixed two-way ANOVA to test the conditioning effect indicated that significant differences occurred for the group (F1, 14 = 43.18, *p* < 0.05), session (F4, 56 = 4.96, *p* < 0.05), and the interaction of group and session (F4, 56 = 15.36, *p* < 0.05). The post hoc Tukey test indicated that significant differences existed in Sessions 2–5 (*p* < 0.05) but not in Session 1 (*p* > 0.05; Figure 4A). For the extinction effect, significant differences occurred for the group (F1, 14 = 20.83, *p* < 0.05), session (F4, 56 = 13.89, *p* < 0.05), and the interaction of group and session (F4, 56 = 13.42, *p* < 0.05). Over Sessions 1, 2, and 4, there were significant differences between the saline and morphine groups (*p* < 0.05). In particular, there was no significant difference between the saline and morphine groups about the intake volume of the saccharin solution in Sessions 3 and 5 during the extinction phase (*p* > 0.05; Figure 4B). Concerning reinstatement, an independent *t*-test indicated that in terms of the mean intake volume of the saccharin solution, morphine had significantly less effect than saline (*t* = 4.22, *p* < 0.05; Figure 4C). Therefore, morphine produced conditioned suppression (Figure 4A). Moreover, morphine exhibited an extinction effect that increased the intake volume of the saccharin solution over the five sessions (Figure 4B). Morphine also induced a reinstatement effect (Figure 4C).

### 3.3. Plasma Corticosterone Levels on Conditioning, Extinction, and Reinstatement

The results indicated that no significant differences between the saline and morphine groups occurred at baseline (F1, 10 = 2.02, *p* > 0.05) and in the extinction phase (F1, 10 = 1.40, *p* > 0.05). The morphine group had significantly higher corticosterone levels in the conditioning (F1, 10 = 84.02, *p* < 0.05) and reinstatement (F1, 10 = 13.92, *p* < 0.05) phases than the saline group did (Figure 5). Morphine-induced conditioned suppression was associated with high plasma corticosterone levels. This effect of high corticosterone levels also occurred in the reinstatement phase. Therefore, the morphine-conditioned suppression of saccharin solution intake was due to an aversive effect rather than a reward. In addition, the reinstatement of morphine, conditioned with a saccharin solution, resulted from aversion rather than a reward.

### 3.4. Which Neural Substrates Mediate CTA by Morphinein Conditioning, Extinction, and Reinstatement?

#### 3.4.1. Conditioning Phase

A one-way ANOVA for c-Fos expression indicated that significant differences occurred for Cg1 (F1, 10 = 22.96, *p* < 0.05), PrL (F1, 10 = 15.74, *p* < 0.05), IL (F1, 10 = 29.72, *p* < 0.05), NAc (F1, 10 = 46.30, *p* < 0.05), BLA (F1, 10 = 50.59, *p* < 0.05), and DG (F1, 10 = 124.64, *p* < 0.05). Nonsignificant differences occurred for CeA (F1, 10 = 0.00, *p* > 0.05), CA1 (F1, 10 = 0.14, *p* > 0.05), CA2 (F1, 10 = 0.17, *p* > 0.05), CA3 (F1, 10 = 1.45, *p* > 0.05), LH (F1, 10 = 1.64, *p* > 0.05), and PC (F1, 10 = 0.00, *p* > 0.05; Figure 6, Figure 7, Figure 8 and Figure 9 and Appendix A). Concerning p-ERK expression onthe conditioning phase, a one-way ANOVA indicated that significant differences occurred for PrL (F1, 10 = 18.22, *p* < 0.05), IL (F1, 10 = 8.91, *p* < 0.05), NAc (F1, 10 = 90.21, *p* < 0.05), and BLA (F1, 10 = 31.92, *p* < 0.05). In the same phase, nonsignificant differences occurred for Cg1 (F1, 10 = 0.20, *p* > 0.05), CeA (F1, 10 = 0.69, *p* > 0.05), CA1 (F1, 10 = 0.09, *p* > 0.05), CA2 (F1, 10 = 0.13, *p* > 0.05), CA3 (F1, 10 = 0.70, *p* > 0.05), DG (F1, 10 = 0.63, *p* > 0.05), LH (F1, 10 = 0.00, *p* > 0.05), and PC (F1, 10 = 0.00, *p* > 0.05; Figure 6, Figure 7, Figure 8 and Figure 9 and Appendix A). In conclusion, Cg1, PrL, IL, NAc, BLA, and DG showed increases in morphine-induced c-Fos expressions during the conditioning phase. PrL, IL, NAc, and BLA showed increases in morphine-induced p-ERK expressions during the conditioning phase.

#### 3.4.2. Extinction Phase

The c-Fos assessments during the extinction phase showed significantly higher levels for PrL (F1, 10 = 5.32, *p* < 0.05), IL (F1, 10 = 44.91, *p* < 0.05), and LH (F1, 10 = 25.23, *p* < 0.05) in the morphine group. However, Cg1 (F1, 10 = 0.07, *p* > 0.05), NAc (F1, 10 = 0.03, *p* > 0.05), CeA (F1, 10 = 0.07, *p* > 0.05), BLA (F1, 10 = 0.01, *p* > 0.05), CA1 (F1, 10 = 0.07, *p* > 0.05), CA2 (F1, 10 = 0.99, *p* > 0.05), CA3 (F1, 10 = 0.73, *p* > 0.05), DG (F1, 10 = 0.01, *p* > 0.05), and PC (F1, 10 = 0.28, *p* > 0.05) did not show any significant differences between the saline and morphine groups (Figure 10, Figure 11, Figure 12 and Figure 13 and Appendix A). In the extinction phase, the p-ERK data showed significantly higher measures for PrL (F1, 10 = 78.57, *p* < 0.05), IL (F1, 10 = 15.74, *p* < 0.05), NAc (F1, 10 = 85.50, *p* < 0.05), LH (F1, 10 = 11.33, *p* < 0.05), and PC (F1, 10 = 17.30, *p* < 0.05) for the morphine group than for the saline group. On the other hand, Cg1 (F1, 10 = 0.30, *p* > 0.05), CeA (F1, 10 = 0.54, *p* > 0.05), BLA (F1, 10 = 0.00, *p* > 0.05), CA1 (F1, 10 = 0.68, *p* > 0.05), CA2 (F1, 10 = 0.29, *p* > 0.05), CA3 (F1, 10 = 0.18, *p* > 0.05), and DG (F1, 10 = 1.00, *p* > 0.05) did not show significant differences between the saline and morphine groups (Figure 10, Figure 11, Figure 12 and Figure 13 and Appendix A). In summary, for c-Fos expression, the PrL, IL, and LH were involved in the extinction effect for the morphine-induced conditioned suppression of saccharin solution intake. Testing p-ERK expression, the PrL, IL, NAc, LH, and PC were involved in the extinction effect for the morphine-induced conditioned suppression of saccharin solution intake.

#### 3.4.3. Reinstatement Phase

In the reinstatement phase, the c-Fos data indicated that PrL (F1, 10 = 14.44, *p* < 0.05), IL (F1, 10 = 22.54, *p* < 0.05), NAc (F1, 10 = 76.50, *p* < 0.05), CeA (F1, 10 = 77.42, *p* < 0.05), BLA (F1, 10 = 23.44, *p* < 0.05), LH (F1, 10 = 32.03, *p* < 0.05), and PC (F1, 10 = 10.29, *p* < 0.05) were more active in the morphine group than in the saline group. The c-Fos expression of Cg1 (F1, 10 = 0.06, *p* > 0.05), CA1 (F1, 10 = 1.44, *p* > 0.05), CA2 (F1, 10 = 0.54, *p* > 0.05), CA3 (F1, 10 = 0.10, *p* > 0.05), and DG (F1, 10 = 0.00, *p* > 0.05) did not indicate a significant difference between the saline and morphine groups (Figure 14, Figure 15, Figure 16 and Figure 17 and Appendix A). In the reinstatement phase, the p-ERK assessment indicated that Cg1 (F1, 10 = 21.86, *p* < 0.05), PrL (F1, 10 = 69.95, *p* < 0.05), IL (F1, 10 = 44.17, *p* < 0.05), NAc (F1, 10 = 59.03, *p* < 0.05), CeA (F1, 10 = 22.06, *p* < 0.05), BLA (F1, 10 = 20.48, *p* < 0.05), and LH (F1, 10 = 32.38, *p* < 0.05) were more active in the morphine group than in the saline group. The p-ERK expressions of CA1 (F1, 10 = 0.00, *p* > 0.05), CA2 (F1, 10 = 1.03, *p* > 0.05), CA3 (F1, 10 = 0.06, *p* > 0.05), DG (F1, 10 = 0.01, *p* > 0.05), and PC (F1, 10 = 0.01, *p* > 0.05) did not indicate significant differences between the saline and morphine groups (Figure 14, Figure 15, Figure 16 and Figure 17 and Appendix A). Altogether, the PrL, IL, NAc, CeA, BLA, LH, and PC mediated morphine-induced reinstatement. The Cg1, PrL, IL, NAc, CeA, BLA, and LH governed morphine-induced reinstatement.

### 3.5. Homeostasis: Assessing c-Fos Expressions on Conditioning, Extinction, and Reinstatement

The c-Fos expression data were analyzed for the selected neural substrates (e.g., Cg1, PrL, IL, NAc, CeA, BLA, CA1, CA2, CA3, DG, LH, and PC) by 3 × 2 × 12 (Phase × Drug × Neural Substrate) mixed three-way ANOVA to examine the status of the brain’s homeostasis. There were significant differences in the factors of phase (F2, 30 = 0.71, *p* > 0.05), drug (F1, 30 = 164.77 *p* < 0.05), and neural substrate (F11, 330 = 272.40, *p* < 0.05). Moreover, significant differences occurred in phase × drug (F2, 30 = 3.41, *p* = 0.05), neural substrate × phase (F22, 330 = 14.76, *p* < 0.05), neural substrate × drug (F11, 330 = 31.27, *p* < 0.05), and neural substrate × drug × phase (F22, 330 = 4.25, *p* < 0.05). Therefore, different phases did not show different c-Fos expressions. Morphine induced hyperactivity in c-Fos expressions for the conditioning, extinction, and reinstatement phases. Different neural substrates exhibited different c-Fos expressions. The interaction of phase, drug, and neural substrates was significantly different.

For further analysis, a 2 × 12 (Drug × Neural Substrate) mixed two-way ANOVA indicated that, on conditioning, significant differences occurred in drug (F1, 10 = 47.78, *p* < 0.05), neural substrate (F11, 110 = 119.02, *p* < 0.05), and drug × neural substrate (F11, 110 = 15.20, *p* < 0.05). One-way ANOVA indicated that significant differences occurred in Cg1, PrL, IL, NAc, BLA, and DG (*p* < 0.05). On extinction, two-way ANOVA indicated that significant differences occurred in drug (F1, 10 = 28.68, *p* < 0.05), neural substrate (F11, 110 = 102.03, *p* < 0.05), and drug × neural substrate (F11, 110 = 11.41, *p* < 0.05). One-way ANOVA indicated that significant differences occurred in PrL, IL, and LH (*p* < 0.05). On reinstatement, two-way ANOVA indicated that significant differences occurred in drug (F1, 10 = 129.35, *p* < 0.05), neural substrate (F11, 110 = 74.16, *p* < 0.05), and drug × neural substrate (F11, 110 = 13.15, *p* < 0.05). One-way ANOVA indicated that significant differences occurred in PrL, IL, NAc, CeA, BLA, LH, and PC (*p* < 0.05; Figure 18).

In addition, a one-way repeated trend analysis was conducted on the conditioning, extinction, and reinstatement phases. In the conditioning phase, the saline group significantly differed in a linear trend (F1, 5 = 0.01, *p* > 0.05) for these determined brain areas. Moreover, the morphine group in the conditioning phase showed a significant difference in an Order 6 trend (F1, 5 = 7.96, *p* < 0.05), but not an Order 7 trend (F1, 5 = 6.10, *p* > 0.05). The c-Fos expressions of the morphine group induced a stronger nonhomeostasis for these 12 neural substrates than those of the saline group during the conditioning phase. The start mark of the neural substrates indicated significant differences for neural hyperactivity on the morphine-induced conditioning suppression of the saccharin solution intake (Figure 18A).

In the extinction phase, the saline group showed a significant difference in an Order 6 trend (F1, 5 = 8.62, *p* < 0.05), but not an Order 7 trend (F1, 5 = 1.28, *p* > 0.05). Moreover, the morphine group revealed a significant difference in an Order 11 trend (F1, 5 = 7.85, *p* < 0.05). The c-Fos expression of morphine during extinction showed partial significance on three specific neural substrates (i.e., PrL, IL, and LH) compared to the saline group. It means that morphine injection also induced less nonhomeostasis on extinction than morphine injection on conditioning (Figure 18B).

In the reinstatement phase, the saline group showed a significant difference in an Order 5 trend (F1, 5 = 84.38, *p* < 0.05), but not an Order 6 trend (F1, 5 = 1.60, *p* > 0.05). The morphine group showed a significant difference in an Order 6 trend (F1, 5 = 71.82, *p* < 0.05), but not an Order 7 trend (F1, 5 = 0.48, *p* > 0.05). The c-Fos expressions of reinstatement induced by morphine injection indicated very severe nonhomeostasis in seven neural substrates, including IL, PrL, BLA, NAc, CeA, LH, and PC when compared to those of conditioning and extinction (Figure 18C).

In summary, to combine ANOVA analysis and trend analysis results, morphine administration seemingly induced nonhomeostasis and disturbance in the conditioning, extinction, and reinstatement phases. The reinstatement phase exhibited the most potent disturbance in the brain, and the extinction phase might exhibit minor disturbance and nonhomeostasis by morphine.

### 3.6. Homeostasis: Assessing p-ERKExpressions on Conditioning, Extinction, and Reinstatement

The p-ERK expression data were also analyzed for the selected brain areas (e.g., Cg1, PrL, IL, NAc, CeA, BLA, CA1, CA2, CA3, DG, LH, and PC) by 3 × 2 × 12 (Phase × DrugNeural Substrate) mixed three-way ANOVA. The results showed significant differences in phase (F2, 30 = 15.98, *p* < 0.05), drug (F1, 30 = 220.23, *p* < 0.05), and neural substrate (F11, 330 = 350.61, *p* < 0.05). Moreover, significant differences also occurred in phase × drug (F2, 30 = 3.93, *p* < 0.05), neural substrate x phase (F22, 330 = 6.06, *p* < 0.05), neural substrate × drug (F11, 330 = 37.79, *p* < 0.05), and neural substrate × drug × phase (F22, 330 = 2.39, *p* < 0.05). Therefore, the phase factor showed a different p-ERK expression. p-ERK expressions of the morphine group were hyperactive in the conditioning, extinction, and reinstatement phases. Different p-ERK expressions occurred in the various brain areas. Phase, drug, and neural substrate showed an interactive effect in p-ERK hyperexpression.

When analyzing p-ERK data further using a 2 × 12 (Drug × Neural Substrate) mixed two-way ANOVA, the results indicated that, on conditioning, significant differences occurred in drug (F1, 10 = 26.18, *p* < 0.05), neural substrate (F11, 110 = 97.79, *p* < 0.05), and drug × neural substrate (F11, 110 = 9.36, *p* < 0.05). One-way ANOVA indicated that significant differences occurred in IL, PrL, BLA, and NAc (*p* < 0.05). On extinction, two-way ANOVA indicated that significant differences occurred in drug (F1, 10 = 125.70, *p* < 0.05), neural substrate (F11, 110 = 123.93, *p* < 0.05), and drug × neural substrate (F11, 110 = 17.77, *p* < 0.05). One-way ANOVA indicated that significant differences occurred in PrL, IL, NAc, LH, and PC (*p* < 0.05). On reinstatement, two-way ANOVA indicated that significant differences occurred in drug (F1, 10 = 198.80, *p* < 0.05), neural substrate (F11, 110 = 172.66, *p* < 0.05), and drug x neural substrate (F11, 110 = 22.01, *p* < 0.05). One-way ANOVA indicated that significant differences occurred in Cg1, PrL, IL, BLA, NAc, CeA, and LH (*p* < 0.05; Figure 19). Table 2 depicts that analysis of the number and percentage of neural substrates for c-Fos or p-ERK expression during conditioning, extinction, and reinstatement.

The p-ERK labeling data were analyzed by a one-way repeated trend analysis for conditioning, extinction, and reinstatement phases for homeostasis testing. In the conditioning phase, the p-ERK data showed a significant difference in an Order 9 trend (F1, 5 = 10.66, *p* < 0.05), but not an Order 10 trend (F1, 5 = 1.30, *p* > 0.05). The morphine group showed a significant difference in an Order 6 trend (F1, 5 = 16.05, *p* < 0.05), but not an Order 7 trend (F1, 5 = 0.02, *p* > 0.05). The p-ERK expression revealed a different pattern of trend between the saline and morphine groups. Although the saline group showed a more robust expression of p-ERK than the morphine group, the PrL, IL, NAc, and BLA showed a higher expression of p-ERK in the morphine group, indicating morphine injections induced disturbance and nonhomeostasis in the selected neural substrates and probably produced neural plasticity in the PrL, IL, NAc, and BLA (Figure 19A).

In the extinction phase, the p-ERK expressions of the saline group showed a significant difference in an Order 5 trend (F1, 5 = 10.95, *p* < 0.05), but not an Order 6 trend (F1, 5 = 1.14, *p* > 0.05). The morphine group showed a significant difference in an Order 6 trend (F1, 5 = 7.38, *p* < 0.05), but not an Order 7 trend (F1, 5 = 2.02, *p* > 0.05). p-ERK expressions of morphine produced more nonhomeostasis status in the extinction phase than that of saline injections, indicating this kind of nonhomeostasis occurred in the PrL, IL, NAc, LH, and PC (Figure 19B).

In the reinstatement phase, the p-ERK expressions of the saline group indicated a significant difference in an Order 5 trend (F1, 5 = 44.45, *p* < 0.05), but not an Order 6 trend (F1, 5 = 4.58, *p* > 0.05). The morphine group showed a significant difference in an Order 6 trend (F1, 5 = 34.07, *p* < 0.05), but not an Order 7 trend (F1, 5 = 0.14, *p* > 0.05). Therefore, morphine injections on reinstatement showed severe nonhomeostasis compared to the saline group, and this kind of nonhomeostasis occurred in the Cg1, PrL, IL, NAc, CeA, BLA, and LH for p-ERK overexpression. The results mean that neural plasticity occurred in the Cg1, PrL, IL, NAc, CeA, BLA, and LH after the morphine-induced conditioned suppression of saccharin solution in thereinstatement phase (Figure 19C).

In summary, regardless of c-Fos and p-ERK expressions, the results have shown that the nonhomeostasis status produces higher neural activity and neural plasticity in the reinstatement phase than that of the conditioning and extinction phases. It means that the subjects produce the most vigorous disturbance and nonhomeostasis in drug relapse, but there was minor disturbance and nonhomeostasis in the extinction and cessation phases.

## 4. Discussion

The present study indicated that morphine doses higher than 20 mg/kg simultaneously produced reward in CPP and aversion in CTA; the lower (or equal to) 10 mg/kg morphine dose only induced aversion in CTA but not reward inCPP (Table 1). Second, the 10 mg/kg morphine dose was determined to induce only an aversive effect of morphine that resulted in conditioned taste suppression in behavior. For c-Fos labeling, in the conditioning phase, the Cg1, IL, PrL, NAc, DG, and BLA were involved in the morphine-induced CTA effect. On extinction, the IL, PrL, and LH, which were demonstrated as aversive neural substrates, played a crucial role in the extinction effect. The Cg1, PrL, IL, NAc, CeA, BLA, LH, and PC were active in morphine-induced CTA reinstatement. In addition, in the conditioning phase, morphine-induced conditioned taste suppression was associated with higher plasma corticosterone levels, indicating that conditioned suppression is due to aversion rather than the reward of abused drugs. Third, concerning the homeostasis analysis using c-Fos and p-ERK, the results have shown that the number and percentage of neural substrates for neural hyperactivity were the highest in the reinstatement phase, indicating the status of nonhomeostasis and disturbance occurred in the reinstatement phase when compared to those of the conditioning and extinction phases. The second highest was in the conditioning phase. The lowest was in the extinction phase. It means that the abusers produced the most potent brain disturbance and nonhomeostasis in drug relapse, but minor disturbance and nonhomeostasis were seen in the extinction and cessation phase (see Figure 18 and Figure 19; Table 2). The present findings are exciting because when morphine was not given on extinction, many neural substrates (e.g., PrL, NAc, and PC) facilitated neural plasticity under the neural activity condition, particularly in the extinction phase. In summary, the extinction and cessation phases are seemingly crucial for neural plasticity; however, this kind of neural plasticity occurs less in the conditioning and reinstatement phases. These findings might provide some implications for novel treatments of drug addiction in further studies.

### 4.1. Debate Issue: Challenging the Reward Comparison Hypothesis

The present study revealed some data that challenge the reward comparison hypothesis that the morphine-induced conditioned taste suppression is due to the reward comparison between the CS taste and the US morphine. First, the higher doses of morphine (≥20 mg/kg) simultaneously induced reward in CPP and aversion in CTA, indicating the abused drug morphine produced the paradoxical effect—reward and aversion. The data supported Riley and colleagues’ findings [1,35,36]. Second, the morphine-induced conditioning effect in behavior cannot be explained by the reward comparison hypothesis. For example, the CS saccharin solution intake measurements without the US agent (morphine injections) revealed an extinction effect. Later, rats were given US morphine injection, and the conditioned suppression of saccharin solution intake was revealed again, showing the reinstatement effect. This phenomenon of extinction and reinstatement followed the rules of the CTA conditioned learning but did not support the reward comparison hypothesis. The extinction and reinstatement processes are difficult to explain using the reward comparison hypothesis. Therefore, morphine-induced conditioned taste suppression is a kind of conditioned taste aversion learning but not a reward comparison between CS and US. This kind of conditioned taste suppression learning induced by morphine is similar to LiCl-induced CTA, particularly to the dose of 10 mg/kg morphine. Third, the data of the neuroendocrinological test related to corticosterone levels did not support the reward comparison hypothesis. For example, our study showed that a significantly higher corticosterone level occurred in conditioning and reinstatement when compared to the saline and morphine groups. However, no significant difference was observed in corticosterone levels between the saline and morphine groups. The results indicated that morphine-induced conditioned taste suppression was due to the aversive effect of morphine for the conditioning and reinstatement phases. This aversive effect of morphine disappeared in extinction. Therefore, the task of conditioned taste suppression induced by abused drugs is very similar to the typical aversion LiCl-induced CTA. Morphine-induced conditioned taste suppression was demonstrated to be a kind of CTA. Abused drugs such as morphine would show the aversive property in this task, and presumably, it revealed the rewarding property in the CPP task. The paradoxical effect of abused drugs can simultaneously show reward and aversion, and the present finding was consistent with the viewpoint of the task-dependent drug effect hypothesis [9]. In addition, the present data were consistent with the previous findings [9,10,11,41,42]. For example, the microinjection of amphetamine into the rewarding NAc induces a CPP effect; however, the microinjection of amphetamine into the aversive area postrema induces a conditioned suppression effect [41]. In one study, rats were trained to run down a runway and then eat food that was paired with morphine. They indicated an increase in running speed and reduced food consumption; morphine exhibited a reward in the runway test and aversion in the CTA [42]. The backward and forward conditioning between the CS and US, regardless of aversive LiCl or rewarding amphetamine, showed the conditioned suppression of CS solution intake. In addition, the synergic injection of rewarding amphetamine and aversive LiCl exhibited a strong conditioned taste suppression effect but did not eliminate the conditioned taste suppression effect [9]. Compared to high doses of ethanol (including 0.35 g/kg and 0.5 g/kg), only the middle dose of ethanol (i.e., 0.20 g/kg) simultaneously revealed an aversion in conditioned taste suppression and a reward in CPP [11]. These data contradict the notion of reward comparison in terms of the reward value of the CS solution and the US, thus challenging the reward comparison hypothesis. In contrast, the data support the notion that abused drugs simultaneously induced a paradoxical effect—reward and aversion—which we named the paradoxical effect hypothesis of abused drugs.

### 4.2. Current Viewpoints of Neural Substrates in Drug Addiction for Reward and Aversion

Reviewing the literature from 1970 to 2019 using the PubMed database, the CPP task is the most popular task to test drug addiction’s reward and reinforcement effects in morphine-induced operant conditioning. A total of 1085 papers have been stored in PubMed on the CPP and morphine in the animal model. Moreover, 21.77% of the research was on all reward and aversion tasks in the animal model. Alternatively, since 1970, 675 papers have been published on the drug self-administration task, and 13.45% were on all reward and aversion tasks in the morphine animal model. Thus, the CPP task has been studied the most to examine the rewarding effect of morphine on conditioned learning (Figure 20A,B). Regarding the studies of drug addiction and neural substrates in morphine use related to reward, since 1970, four papers have been published on the drug self-administration task, and seven have been published on the CPP task (Figure 20C).

In contrast to the topic of reward and drug addiction, the aversion or aversive conditioning of drug addiction is oppositely much less studied except in the research related to withdrawal symptoms. This line of the study of aversion and aversive learning of drug addiction has often been carried out to test the withdrawal symptoms of the animal model for opiate addiction (Figure 20A). A total of 2840 papers have been published on aversive withdrawal symptoms of morphine addiction, and 56.96% of the studies were on the reward and aversion of morphine addiction (Figure 20B). Eighteen papers have been published on neural substrates involved in aversive drug addiction in morphine’s withdrawal symptoms, indicating most of the research is related to drug addiction and neural substrates (Figure 20C). Less research has been carried out on aversive conditioning for conditioned place aversion (CPA) and CTA related to morphine addiction except for withdrawal symptoms. Only 268 and 117 papers, respectively, were on CPA and the CTA (Figure 20A). The CPA topic was 5.38% of all reward and aversion morphine addiction studies in the animal model, and CTA was 2.35% (Figure 20B).

Furthermore, only seven papers researched CPA in neural substrates for morphine addiction, whereas those of CTA did not have any research to investigate the issue of neural substrates involved in morphine-induced CTA (Figure 20C). An abused drugs-induced CTA task is seldom used to examine the aversion or aversive learning of drug addiction. It is probably due to the previous paradoxical issue of the conditioned taste suppression induced by abused drugs resulting from the reward or aversion of abused drugs. Thus, the reward comparison hypothesis and the conventional CTA viewpoint remain to be debated. Based on the present data in behavior and neuroendocrinological corticosterone tests, we can be confident when suggesting that the conditioned taste suppression is induced by morphine due to aversion but not reward. The present study is the first to use the morphine-induced CTA task and comprehensively examine which neural substrates are involved in the aversion to morphine in the conditioning, extinction, and reinstatement phases.

### 4.3. mPFC: Cg1, PrL, and IL

Depending on the previous findings, the subareas of the mPFC (including the Cg1, PrL, and IL) regulate the rewarding effect or reinforcement but not aversion in drug addiction [43]. For example, the anterior cingulate cortex, overlapped in the Cg1, was associated with the reward-related signal and stimulus in addiction behaviors [19]. In a recent study, the PrL was shown to govern cocaine-seeking behavior, and the IL was shown to play a role in inhibiting cocaine-seeking behavior in the extinction phase [27]. The GABA neurons of the PrL and IL have been shown to mediate cocaine-seeking behavior in a discriminative stimulus self-administration task [17]. A recent optogenetic study demonstrated that the optical inhibition to the pyramidal neurons of the IL enhanced active lever pressing on the shortened extinction but did not affect the retention of extinction learning, indicating the IL is involved in extinction in the cocaine self-administration task [44]. Therefore, the mPFC (including Cg1, PrL, and IL) is probably involved in the reward effect in drug addiction.

The present results showed that the PrL and IL were hyperactive in c-Fos and p-ERK expressions in all three phases: conditioning, extinction, and reinstatement; the Cg1 was shown to have c-Fos and p-ERK expression on conditioning and reinstatement, respectively (Figure 21). These results cannot be explained with the reward comparison hypothesis, as the Cg1, PrL, and IL of the mPFC involved in the aversion resulted in morphine-induced conditioned taste suppression. Therefore, based on our data, the subregions of the mPFC, such as Cg1, PrL, and IL mediate the aversive effect of abused drugs but not reward. Our results did not support the conventional view that the reward or reinforcement was controlled by the Cg1, PrL, and IL of the mPFC in drug addiction. The issue of whether the mPFC (e.g., Cg1, PrL, and IL) is involved in the paradoxical effect—reward and aversion in drug addiction—remains to be investigated in further studies.

### 4.4. Hippocampus: CA1, CA2, CA3, and DG

Previous researchers have shown that the hippocampus can control reward-associated addictive behaviors, particularly contextual conditioned learning. For example, the DA and CA1 of the ventral hippocampus exhibited high dendritic spine density and high mRNA levels of BDNF and TrkB for morphine-induced CPP [13]. The subregions of the hippocampus (including CA1, CA2, and CA3) contribute to the expressions of Zif 268 and Fos B in cocaine memory reconsolidation in the CPP task, indicating the CA1, CA2, and CA3 are involved in the rewarding effect of CPP induced by cocaine in drug addiction [45]. The tyrosine kinase B receptor expression, a neurotrophic factor receptor related to synaptic plasticity in the BLA, DG, CA1, and CA3, was shown to regulate amphetamine-induced rewarding effects in the CPP task [28]. As above, the DG of the hippocampus mediated the rewarding effect of drug addiction. A recent study indicated that Arc and early growth response 1 expression in the DG dopamine, glutamate, and GABA neurons were associated with morphine-induced withdrawal-contextual memory reconsolidation in the CPA task, indicating the DG of the hippocampus governed the aversive withdrawal effect induced by morphine to be conditioned with the context to form the CPA [26]. Therefore, the CA1, CA2, and CA3 of the hippocampus were only involved in the rewarding process in drug addiction, and the DG mediates the paradoxical effect—reward and aversion—in drug addiction. The present results revealed that the c-Fos and p-ERK expressions of CA1, CA2, and CA3 did not significantly differ between the saline and morphine groups in any of the conditioning, extinction, and reinstatement phases. Only the DG in the morphine and conditioning phases showed hyperactivity of c-Fos (but not p-ERK) expressions in the morphine-induced aversive CTA (Figure 21). This result for the DG c-Fos hyperexpression supports the previous data regarding the involvement of the DG of the hippocampus in aversion-related addictive behaviors.

### 4.5. Amygdala: CeA and BLA

The subdivision of the amygdala is dissociated from the CeA and the BLA, and these brain subareas play different roles in drug addiction. Conventionally, the CeA is involved in aversive withdrawal symptoms [33,34]. For example, a previous study showed that the c-Fos expressions of the CeA were associated with the acquisition of CPA induced by the morphine withdrawal effect; however, the CeA did not mediate the expressions of the morphine-induced withdrawal effect in CPA [33]. An optogenetic study showed that photoinhibition in the CeA could suppress morphine’s withdrawal symptoms, including aversive, anxiety, and anhedonia behaviors [34]. However, the excitatory optogenetic stimulations in the CeA promoted morphine-induced withdrawal behaviors [34]. Therefore, the CeA contributed to the aversive effects of morphine, particularly withdrawal symptoms. The present results suggested that the c-Fos or the p-ERK expressions of the CeA only occurred in the reinstatement phase, supporting the aversive effect of morphine-induced CTA in reinstatement, which was parallel to former findings of CeA involvement in aversive withdrawal symptoms (Figure 21).

The BLA was also demonstrated to mediate aversive and anxiety behaviors and morphine-induced withdrawal symptoms [46,47]. For example, corticotropin-releasing factor 1 receptor interference was shown to reduce the relapse of drug-seeking behavior induced by opiate withdrawal-associated aversive memory in CPA [46]. NMDA receptor antagonist AP5 injections into the BLA or dorsal hippocampus—but not the CeA—impaired the CPA effect on extinction and reduced the ERK and CREB phosphorylation, indicating BLA involvement in aversive CPA and withdrawal behaviors [47]. Some studies have recently demonstrated that the BLA regulates reward and aversion in drug addiction [32,48]. The microinfusions of the NMDA antagonist into the NA shell—but not the NA core—attenuated the intra-BLA CB1-mediated reward effects and CB1 activation induced by aversive effects of morphine; the BLA CB1 activations regulated the electrophysiological activity in the NA shell, which was associated with the reward and aversion induced by morphine [32]. A protein-signaling study of the BLA indicated that blunts of the BLA D3 receptor prevented the rewarding effect in morphine-induced CPP and the aversive effect in morphine-induced aversive withdrawal memory; the D3 receptor deactivations within the BLA increased Cdk5 phosphorylation and calcineurin expressions, indicating the BLA D3 receptors controlled the opiate-rewarding process and aversion induced by chronic opiate treatments [48]. Therefore, the BLA might possess a paradoxical effect of reward and aversion in drug addiction. The study results suggest that the c-Fos and p-ERK expressions of the BLA were both hyperactive in morphine-induced CTA on conditioning and reinstatement but not extinction, suggesting the involvement of morphine-induced aversion (Figure 21).

### 4.6. NAc

According to the earliest studies, the NAc was potentially involved in the rewarding effect in drug addiction [49]. Previously, some research has proposed the aversive effect of drug addiction through the opponent mechanism in aversive withdrawal symptoms [50]. However, this line of the study did not emphasize the paradoxical effect of reward and aversion. Recently, the NAc was found to regulate the aversive effect of abused drugs [51]. For example, using the CPA and withdrawal models, the microinjections of the protein kinases inhibitor H7 or H8 into the NAc reduced the CPA effect and withdrawal symptoms, suggesting that protein kinases within the NAc are involved therein [52].

Moreover, injections of the D1 receptor antagonist SCH39166 into the NAc abolished amphetamine-induced conditioned aversive suppression [25]. A recent review has reported that the paraventricular nucleus of the hypothalamus and NAc pathway also regulated the aversive withdrawal symptoms of opiates [51]. Furthermore, the thalamus-NAc projection mediated aversive opiate dependence in the CPA and withdrawal symptoms [53].

In summary, the present results indicate that the NAc was activated in c-Fos expressions on conditioning and reinstatement and p-ERK expressions on conditioning, extinction, and reinstatement, suggesting the aversive effect of morphine occurs in the conditioning, extinction, and reinstatement phases (Figure 21). Our data on morphine-induced aversive CTA were consistent with the above view of the NAc’s involvement in the aversive effect of abused drugs, collectively suggesting that the NAc might be involved in the paradoxical effects of morphine-induced reward and aversion.

### 4.7. LH

Another rewarding neural substrate, the LH, showed c-Fos or p-ERK hyperexpression in the extinction and reinstatement phases—but not the conditioning phase—of this study (Figure 21). Present results on the LH’s involvement in the aversive effect of morphine-induced CTA are not consistent with those of previous studies, in which the LH mediated reward processing and drug addiction [16,29,54]. For example, a recent review reported that the orexin-1 and -2 receptor antagonists blunt the rewarding conditioning, indicating that orexin neurons of the lateral hypothalamus are involved in drug addiction [29]. Electroacupuncture could potentially interfere with the CPP and reinstatement of blunting CPP induced by morphine, while LH orexin neurons mediated the drug-seeking behavior [54]. Furthermore, the orexin signaling enhanced the synaptic plasticity of glutamatergic neurons of the lateral hypothalamus onto dopamine neurons in the ventral tegmental area [16]. According to our data and previous evidence, the LH is not involved in the rewarding effect of abused drugs and has an essential role in the aversive effect of morphine-induced CTA; the role of the LH in drug addiction should therefore be further reviewed. The LH seemingly mediates reward as well as aversion in many studies of drug addiction. Whether the LH is involved in the paradoxical effect of reward and aversion remains to be seen in further studies.

### 4.8. PC

The results for the PC show significant hyperactivity in c-Fos on reinstatement and in p-ERK expressions on extinction for morphine-induced conditioned taste suppression, indicating the possible aversive effect of morphine in extinction or reinstatement. The present data did not support existing data on motivated behavior in addiction [14,30]. Chronic treatments of amphetamine could induce stereotyped behavior and enhanced BNDF mRNA in the BLA, rostral PC, and paraventricular nucleus of the hypothalamus, indicating the rostral PC, the BLA and the paraventricular hypothalamus contribute to the rewarding process in drug addiction [14]. The PC was found to increase CaMKII-T286 phosphorylation expressions associated with cue-reinstatement of alcohol-seeking behavior, suggesting that the PC might regulate drug addiction’s reward effect [30]. There have been relatively few studies on the role of the PC in reward or reinforcement in drug addiction. However, the present study is the first to investigate the aversion of drug addiction for the PC and its potential role in the paradoxical effect, or reward and aversion in drug addiction (Figure 21). This issue also merits further investigation.

### 4.9. The Paradoxical Effect Hypothesis of Abused Drugs: Nonhomeostasis and Disturbance in the Brain

Riley and colleagues reported that abused drugs might have rewarding and aversive effects [35]. The reward can induce approaching behavior, while aversion can elicit avoidance behavior; for drug abusers, reward-approaching behavior should be weighed alongside aversion-approaching behavior to determine the strength of compulsive behavior [1,35,36,55]. In their viewpoint on drug addiction, researchers have emphasized the importance of balance for approaching responses induced by reward and avoidance behaviors induced by aversion [1], as the aversion of the abused drug has an essential role in drug addiction along with the reward [1]. Conventionally, the reward and aversion of abused drugs have often been termed the paradoxical effect. Riley and colleagues suggested the reward and aversion are not a paradox, and the “paradoxical” effect is attributable to the multiple effects of abused drugs [36]. However, there has been no research to investigate the neural substrates involved in the paradoxical effect; the present study is the first one to address this issue.

In comparing our data with previous evidence, we found that our results were consistent with the paradoxical viewpoint of Riley et al. in drug addiction [1,35,36,55]. Furthermore, after analyzing the immunohistochemical staining data of the neural substrates, our results suggest that an addictive brain reacts to morphine injections with hyperactivity showing in c-Fos or p-ERK expressions, such as the Cg1, PrL, IL, BLA, NAc, and DG, in the morphine-induced CTA. These brain areas, such as Cg1, PrL, and IL, have previously been involved in reward-conditioned learning of drug addiction [19,43]. Accordingly, both our results and previous data on the Cg1, PrL, and IL of the mPFC suggest that morphine might simultaneously produce the reward and aversion effects in drug addiction, as well as the DG, NAc, and BLA. Therefore, upon conditioning to morphine administrations, the abuser’s brain reveals nonhomeostasis, disturbance, and simultaneous rewarding and aversive activity in these specific neural substrates. It can be assumed that drug abusers conditioned with morphine injections exhibit euphoric feelings and appear to experience aversion or stressful feelings. Furthermore, we provide the paradoxical effect hypothesis of abused drugs to focus on the importance of maintaining homeostasis in the brain. Further studies should address how to achieve the brain’s homeostasis with pharmacological or nonpharmacological strategies.

In comparing our data with previous evidence, we found that our results were consistent with the paradoxical viewpoint of Riley et al. in drug addiction [19,20,21,46]. Furthermore, after analyzing the immunohistochemical staining data of the neural substrates, our results suggest that an addictive brain reacts to morphine injections with hyperactivity showing in c-Fos or p-ERK expressions, such as the Cg1, PrL, IL, BLA, NAc, and DG, in the morphine-induced CTA. These brain areas, such as Cg1, PrL, and IL, have previously been involved in reward-conditioned learning of drug addiction [11,35]. Accordingly, both our results and previous data on the Cg1, PrL, and IL of the mPFC suggest that morphine might simultaneously produce the reward and aversion effects in drug addiction, as well as the DG, NAc, and BLA. Therefore, upon conditioning to morphine administrations, the abuser’s brain reveals nonhomeostasis, disturbance, and simultaneous rewarding and aversive activity in these specific neural substrates. It can be assumed that drug abusers conditioned with morphine injections exhibit euphoric feelings and appear to experience aversion or stressful feelings. Furthermore, we provide the paradoxical effect hypothesis of abused drugs to focus on the importance of maintaining homeostasis in the brain. Further studies should address how to achieve the brain’s homeostasis with pharmacological or nonpharmacological strategies.

### 4.10. Further Studies

Recently, research has shown that the mPFC [56,57] and CeA [58] contribute to reward and aversion in drug addiction. For example, previous studies demonstrated that the subarea of the mPFC, such as the IL, regulated the inhibition of drug-seeking behavior on extinction [17,27]; however, a recent study has shown that the mPFC also mediated the aversive effects, and the mPFC probably caused the paradoxical effect of reward and aversion in drug addiction [57]. In this study, microinjections of the PKMzeta inhibitor ZIP into the IL (but not PrL) interfered with expressions of the morphine-induced CPP and CPA on extinction, indicating that the IL governed the rewarding effect in CPP and the aversive effect in CPA [57]. In addition, the PrL of the mPFC was recently noted to mediate the reward and aversion effects induced by morphine [56]. Changes in the CB1 activation within the PrL modulate opiates’ motivational valence; moreover, CB1 transmission within the PrL alters morphine’s rewards toward aversion, indicating that CB1 controls the rewards of the PrL via a mu-opioid receptor–reward pathway and that a kappa-opioid receptor mediates the morphine-induced aversion pathway [56]. This evidence might explain why the higher doses of morphine could trigger the rewarding effect in CPP and the aversive effect in CTA. The high dose of morphine may simultaneously activate the mu-opioid receptor pathway to produce the rewarding effect in CPP and trigger the kappa-opioid receptor mechanism to induce the aversive effect in CTA. This issue of morphine’s aversive and rewarding mechanisms should be examined in further studies.

On the other hand, the CeA plays an essential role in the aversion of abused drugs, withdrawal symptoms, or acquisition of the CPA controlled by the CeA [33]. Recently, an optogenetic study has shown that the ChR2 optical stimulation in the CeA increased cocaine self-administration. However, this ChR2 motivational effect was disrupted by muscimol/baclofen microinjections or the inhibitory optogenetichalorhodopsin [58]. Moreover, the GABAergic projection from the anterior insular cortex to the CeA mediated a rewarding process in reinstatement, indicating the GABAergic neurons between the anterior insular cortex and the CeA were involved in the relapse [59]. This optogenetic study suggested that the CeA did not only mediate the aversion in addiction, as suggested by the conventional viewpoint; the CeA is also involved in reward conditioning.

Previous studies have demonstrated that the periaqueductal gray matter (PAG) was also involved in conditioned aversive learning [60,61] and aversive behaviors induced by morphine or opiate drugs. For example, the PAG lesion interfered with morphine-induced CTA [60]. Microinjections of the mu-opioid receptor agonist morphine or the kappa-opioid receptor agonist U-50488H in the dorsal PAG caused the CPA effect, indicating the mu- or kappa-opioid receptors mediated the CPA learning [61]. In an elevated plus-maze test, morphine microinjections (such asbenzodiazepine compound, midazolam) in the dorsal PAG increased entries and spent time in the open arm, and morphine-induced anti-anxiety/anti-aversive effect was reversed by a systematic injection of mu-opioid antagonist naloxone [62]. Therefore, the role of PAG was seemingly to mediate morphine-induced aversive conditioning and aversive behaviors.

In conclusion, the issue of whether the mPFC, CeA, and PAG are involved in a single role in reward or aversion, or if these brain areas mediate both reward and aversion effects, remains to be scrutinized in further studies. Moreover, whether all neural substrates of the brain are simultaneously involved in the paradoxical effect of reward and aversion should be further clarified.

### 4.11. Clinical Implication

Investigations of brain mechanisms related to the paradoxical effect of abused drugs could provide new insights or contribute to new interventions for use in drug addiction clinics. The present findings may indicate that hyperactivity of the brain is distributed in many places, resulting in disturbance and nonhomeostasis and involving various rewarding and aversive neural substrates. Drug abusers experience euphoria or happiness from the rewarding property of abused drugs and suffer the stressful or aversive effect. Drug addiction seems to be a complicated process and involves paradoxical effects of different neural substrates in reward and aversion. However, these paradoxical effects of reward and aversion cannot cancel each other out within the brain. The abuser might experience distress and mental disturbance during morphine treatments. Based on present data, recovering the homeostasis of the brain is essential, as is developing novel pharmacological and nonpharmacological interventions to ameliorate the symptoms of drug addiction effectively. Insight from these findings applies in the clinical treatment of drug addiction.

## 5. Conclusions

According to the present data, morphine-induced conditioned taste suppression could simultaneously activate aversive neural substrates. The reward comparison hypothesis cannot account for the present results; in its place, we offer the paradoxical effect hypothesis of abused drugs to explain the existent and our data. The present data of abused drugs-induced paradoxical effects and the brain’s nonhomeostasis and disturbance in hyperactivations of aversive and rewarding neural substrates might develop novel treatments for drug addiction and dependence. Developing pharmacological or nonpharmacological interventions ameliorate the brain’s nonhomeostasis and disturbance to cure drug addiction and dependence is an essential issue for further studies.

## Figures and Tables

**Figure 1 jcm-10-03197-f001:**
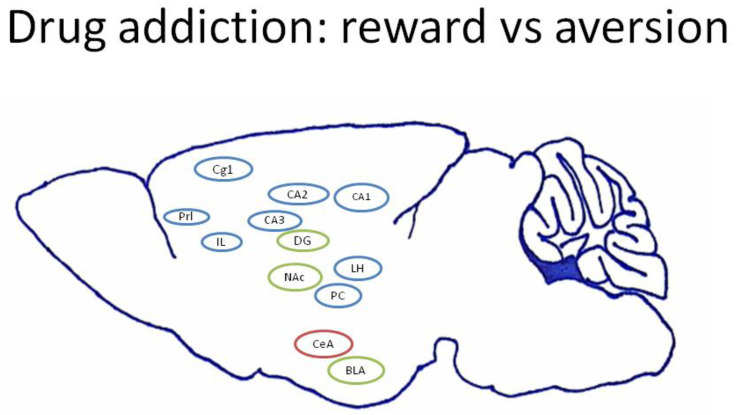
Representation of previous findings regarding neural substrates involved in reward and aversion during drug addiction. The selected neural substrates include the rewarding neural substrates, such as the cingulate cortex area 1 (Cg1), the prelimbic cortex (PrL), the infralimbic cortex (IL), the CA1, the CA2, the CA3, the lateral hypothalamus (LH), and the piriform cortex (PC), as well as the aversive neural substrates, such as the central amygdala (CeA). Other rewarding and aversive neural substrates include the nucleus accumbens (NAc), the basolateral amygdala (BLA), and the dentate gyrus (DG). Note: the rewarding, aversive, or both properties of neural substrates are, respectively, shown in the blue, red, and green circles.

**Figure 2 jcm-10-03197-f002:**
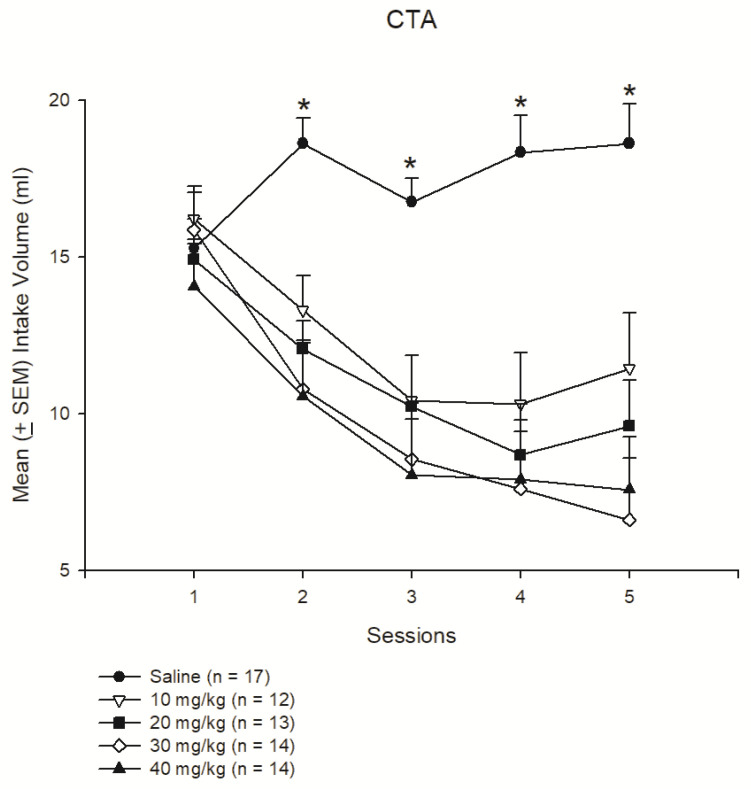
Morphine was acting as an unconditioned stimulus agent in conditioned suppression of intake of a 0.1% saccharin solution. Mean (±SEM) intake volume of 0.1% saccharin solution in rats injected with saline (*n* = 17) or with morphine in doses of 10 mg/kg (*n* = 12), 20 mg/kg (*n* = 13), 30 mg/kg (*n* = 14), and 40 mg/kg (*n* = 14) over 5 sessions. * *p* < 0.05 when comparing the saline groups and the various morphine groups.

**Figure 3 jcm-10-03197-f003:**
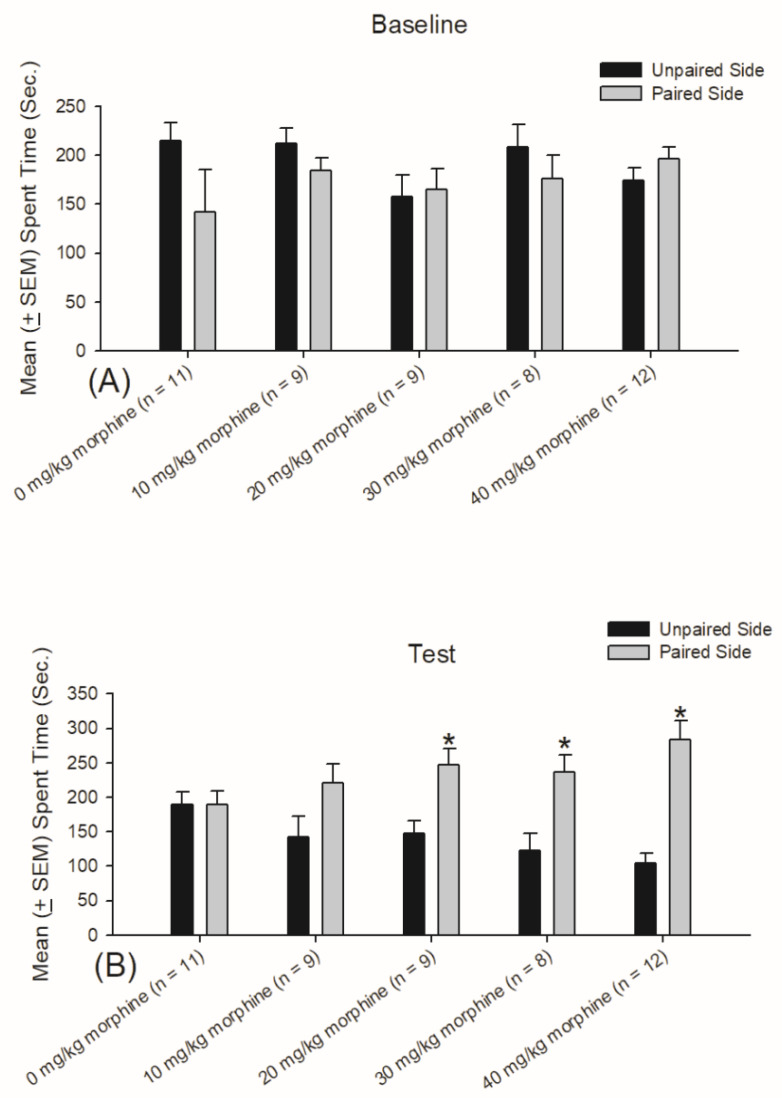
Morphine was acting as the rewarding US in CPP. Mean (±SEM) time spent on saline (*n* = 11), 10 mg/kg morphine (*n* = 9), 20 mg/kg morphine (*n* = 9), 30 mg/kg morphine (*n* = 8), and 40 mg/kg morphine (*n* = 12) groups for the drug-unpaired side and the drug-paired side on the baseline (**A**) and test (**B**) sessions. Note that the drug-paired side means that morphine was injected in one compartment of the CPP box, whereas the drug-unpaired side was the saline injection in the other compartment of the CPP box. * *p* < 0.05 when comparing pre-conditioning and post-conditioning.

**Figure 4 jcm-10-03197-f004:**
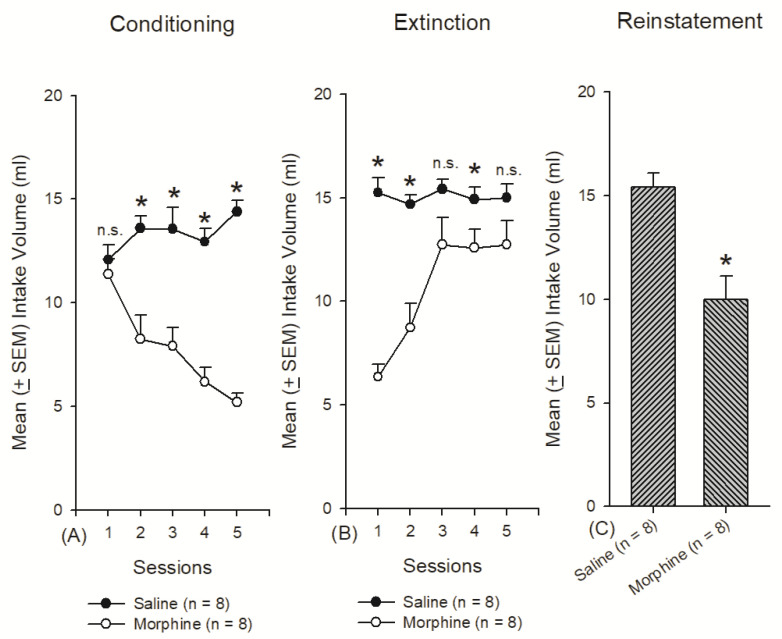
Mean (±SEM) consumption of 0.1% saccharin solution (**A**) for 5 sessions in the conditioning phase (with 10 mg/kg of morphine intraperitoneal injections), (**B**) for 5 sessions in the extinction phase (without morphine injections), and (**C**) for the reinstatement test (with a 10 mg/kg morphine injection) across the saline and morphine groups (*n* = 8 per group). * *p* < 0.05 when comparing the saline and morphine groups.

**Figure 5 jcm-10-03197-f005:**
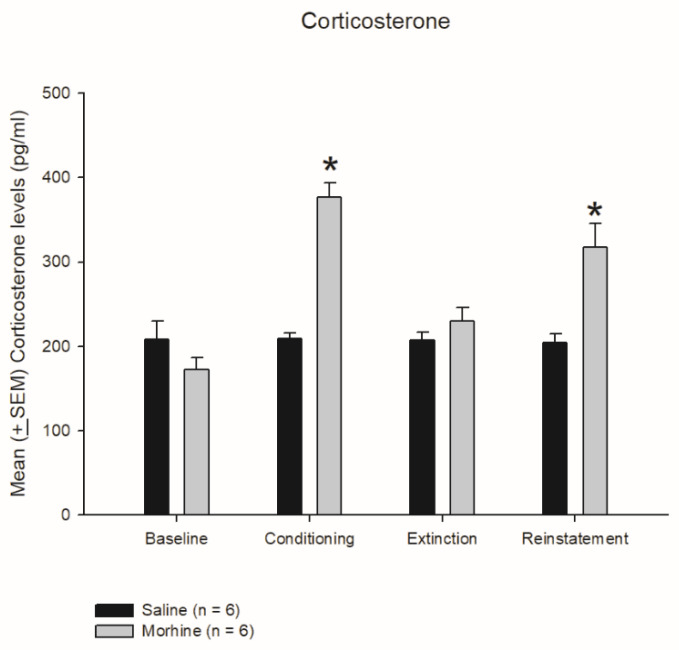
Mean (±SEM) plasma corticosterone levels (pg/mL) in the saline and morphine groups (*n* = 6 per group) at baseline and in the conditioning, extinction, and reinstatement phases. * *p* < 0.05 when comparing the saline and morphine groups (*n* = 6 per group).

**Figure 6 jcm-10-03197-f006:**
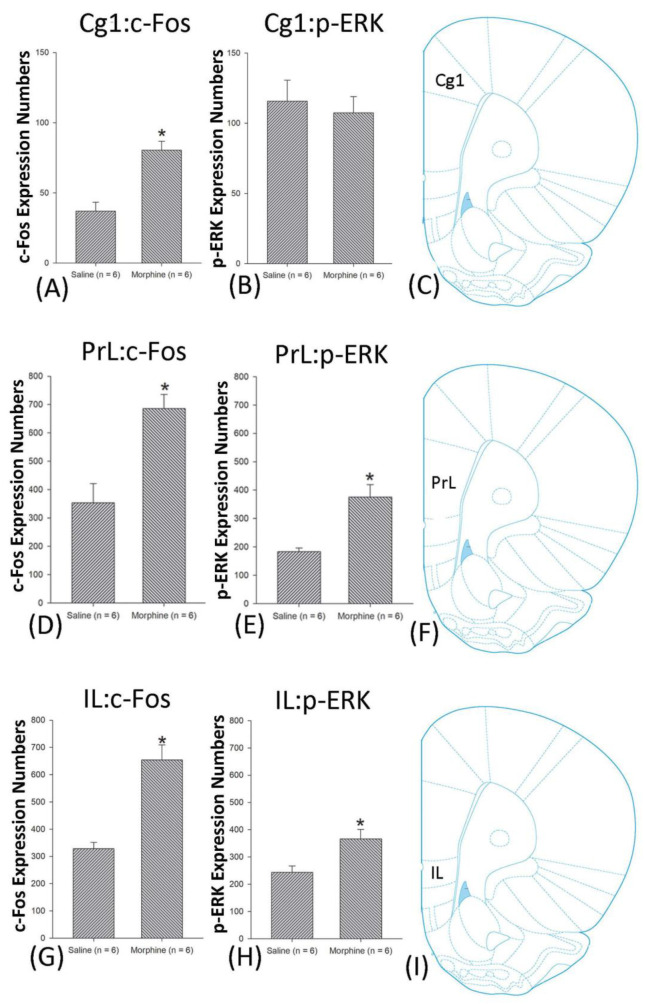
c-Fos or p-ERK expression in the cingulated cortex area 1 (Cg1), prelimbic cortex (PrL) and infralimbic cortex (IL) in conditioning. (**A**) c-Fos and (**B**) p-ERK expression in the Cg1, (**D**) c-Fos and (**E**) p-ERK expression in the PrL, and (**G**) c-Fos and (**H**) p-ERK expression in the IL for conditioning. (**C**,**F**,**I** show a schematic representation of the Cg1, PrL, and IL. * *p* < 0.05 when comparing the saline and morphinegroups (*n* = 6 per group).

**Figure 7 jcm-10-03197-f007:**
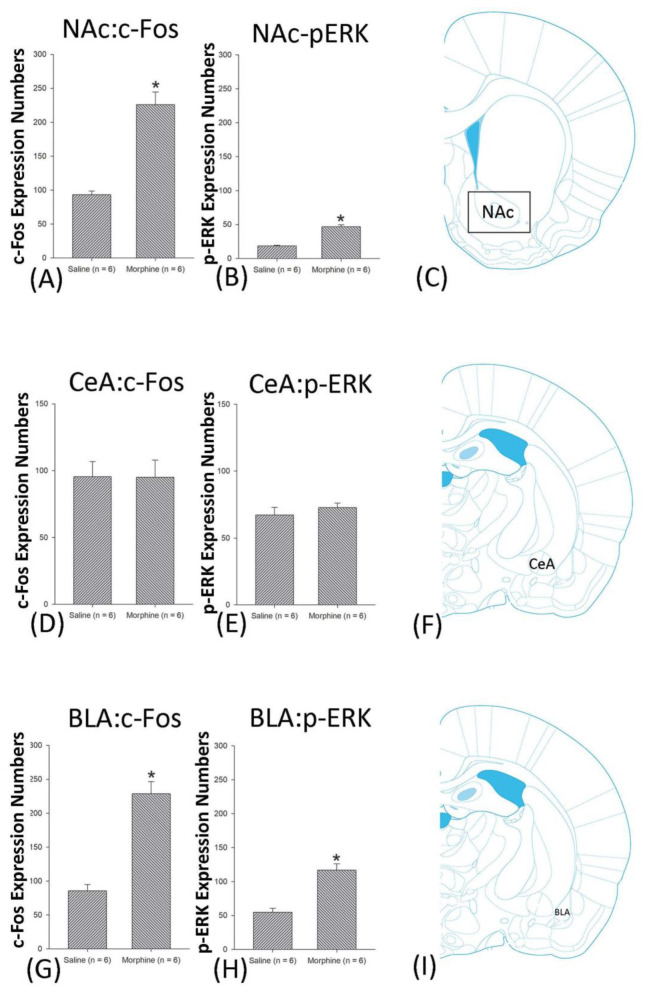
c-Fos or p-ERKimmunoreactivity in the nucleus accumbens (NAc), central amygdala (CeA), and basolateral amygdala (BLA) in conditioning. (**A**) c-Fos expression and (**B**) p-ERK expression in the NAc, (**D**) c-Fos and (**E**). p-ERK expression in the CeA, and (**G**) c-Fos and (**H**). p-ERK expression in the BLA for conditioning. (**C**,**F**,**I**) show a schematic representation of the NAc, CeA, and BLA. * *p* < 0.05 when comparing the saline and morphine groups (*n* = 6 per group).

**Figure 8 jcm-10-03197-f008:**
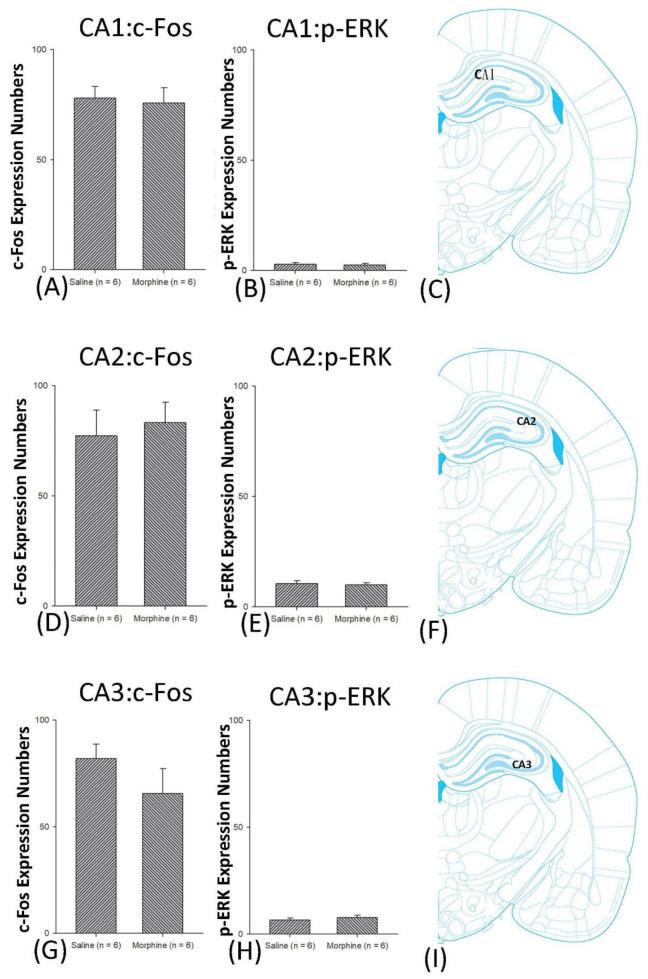
c-Fos or p-ERKimmunoreactivity in the CA1, CA2, and CA3 in conditioning. (**A**) c-Fos expression and (**B**) p-ERK expression in the CA1, (**D**) c-Fos and (**E**). p-ERK expression in the CA2, and (**G**) c-Fos and (**H**) p-ERK expression in the CA3 for conditioning. (**C**,**F**,**I**) show a schematic representation of the CA1, CA2, and CA3. * *p* < 0.05 when comparing the saline and morphine groups (*n* = 6 per group).

**Figure 9 jcm-10-03197-f009:**
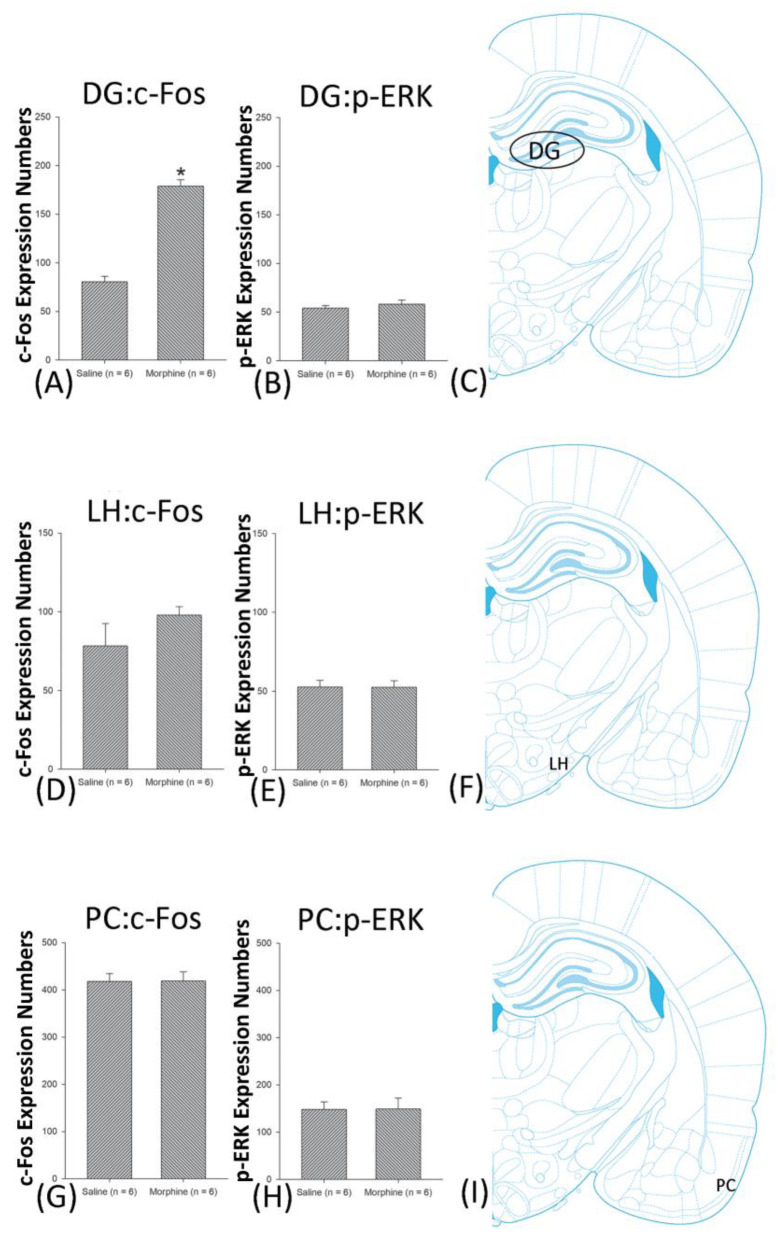
c-Fos or p-ERK expression in the dentate gyrus (DG), lateral hypothalamus (LH), and piriform cortex (PC) in conditioning. (**A**,**B**) c-Fos expression and p-ERK expression in the DG, (**D**,**E**) in the LH, and (**G**,**H**) in PC for conditioning. (**C**,**F**,**I**) show a schematic representation of the DG, LH, and PC. * *p* < 0.05 when comparing the saline and morphine groups (*n* = 6 per group).

**Figure 10 jcm-10-03197-f010:**
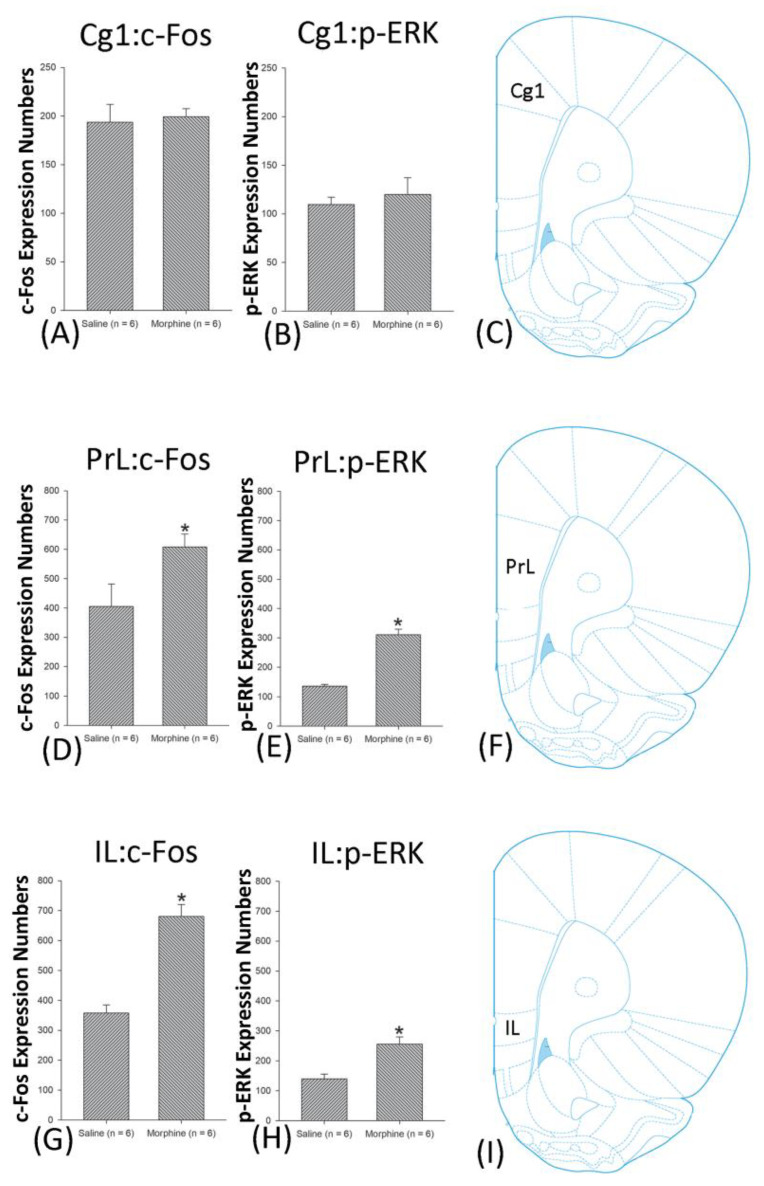
c-Fos and p-ERK expression in the extinction phase for the cingulate cortex area 1 (Cg1), prelimbic cortex (PrL), and infralimbic cortex (IL). (**A**) c-Fos and (**B**) p-ERK expression in the Cg1, (**D**) c-Fos and (**E**) p-ERK expression in the PrL, and (**G**) c-Fos and (**H**) p-ERK expression in the IL for extinction. (**C**,**F**,**I**) indicate a schematic representation of the Cg1, PrL, and IL. * *p* < 0.05 when comparing the saline and morphine groups (*n* = 6 per group).

**Figure 11 jcm-10-03197-f011:**
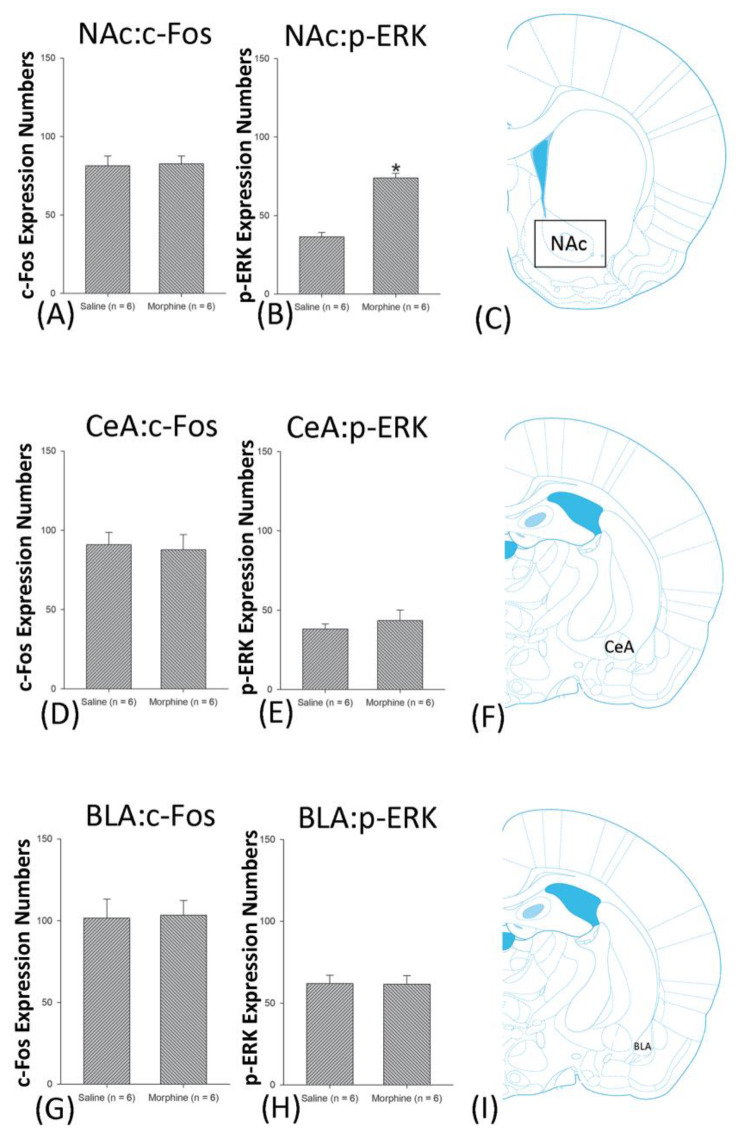
Nucleus accumbens (NAc), central amygdala (CeA), and basolateral amygdala (BLA) express c-Fos or p-ERK proteins in extinction. (**A**) c-Fos and (**B**) p-ERK expression in NAc, (**D**) c-Fos and (**E**) p-ERK expression in the CeA, and (**G**) c-Fos and (**H**) p-ERK expression in the BLA for extinction. (**C**,**F**,**I**) indicate a schematic representation of the NAc, CeA, and BLA. * *p* < 0.05 when comparing the saline and morphine groups (*n* = 6 per group).

**Figure 12 jcm-10-03197-f012:**
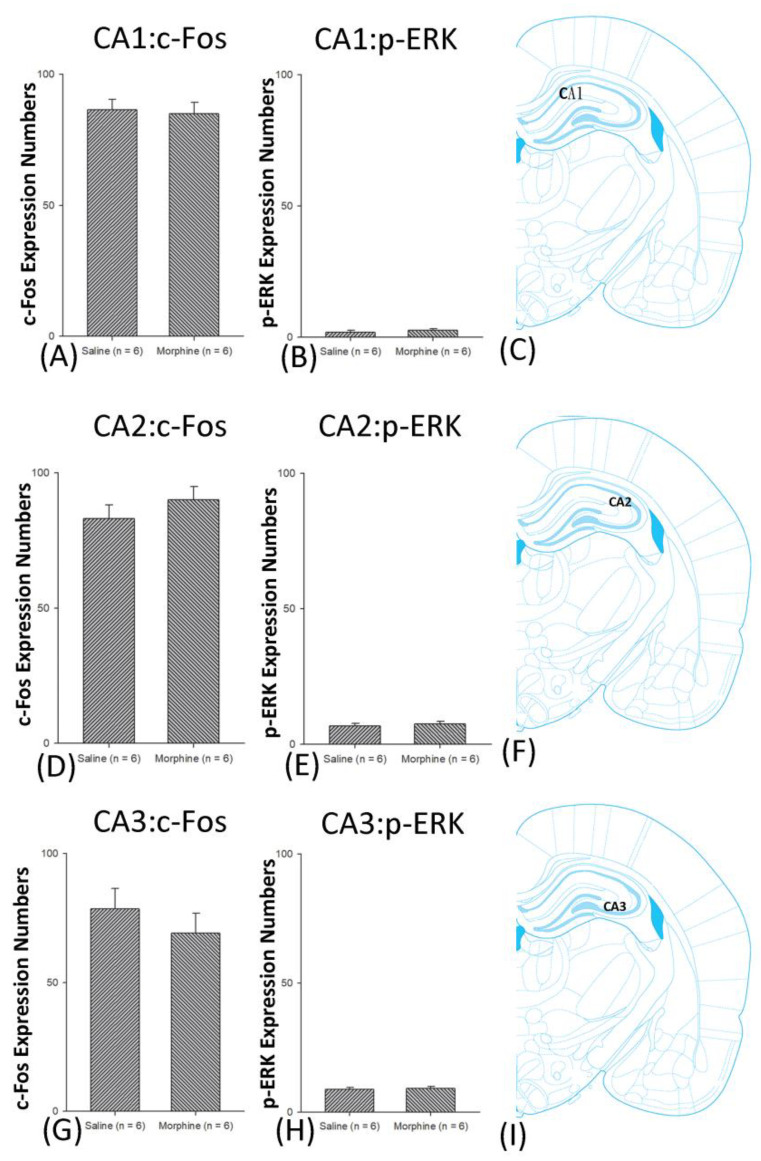
c-Fos or p-ERK immunoreactivity in the CA1, CA2, and CA3 in conditioning. (**A**) c-Fos expression and (**B**) p-ERK expression in the CA1, (**D**) c-Fos and (**E**). p-ERK expression in the CA2, and (**G**) c-Fos and (**H**) p-ERK expression in the CA3 for extinction. (**C**,**F**,**I**) indicate a schematic representation of the CA1, CA2, and CA3. * *p* < 0.05 when comparing the saline and morphine groups (*n* = 6 per group).

**Figure 13 jcm-10-03197-f013:**
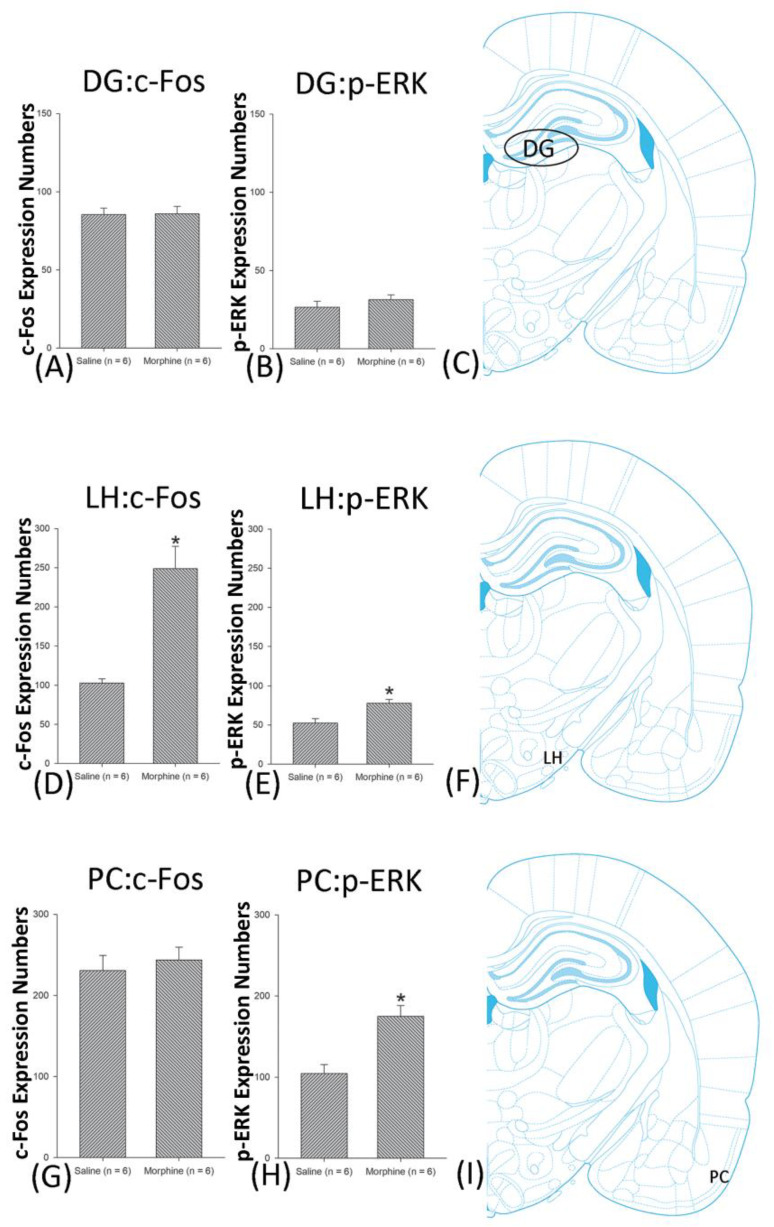
c-Fos or p-ERK expression in the dentate gyrus (DG), the lateral hypothalamus (LH), and the piriform cortex (PC) in extinction. (**A**) c-Fos and (**B**) p-ERK expression in the DG, (**D**) c-Fos and (**E**) p-ERK expression in the LH, and (**G**) c-Fos and (**H**). p-ERK expression in the PC for extinction. (**C**,**F**,**I**) indicate a schematic representation of the DG, LH, and PC. * *p* < 0.05 when comparing the saline and morphine groups (*n* = 6 per group).

**Figure 14 jcm-10-03197-f014:**
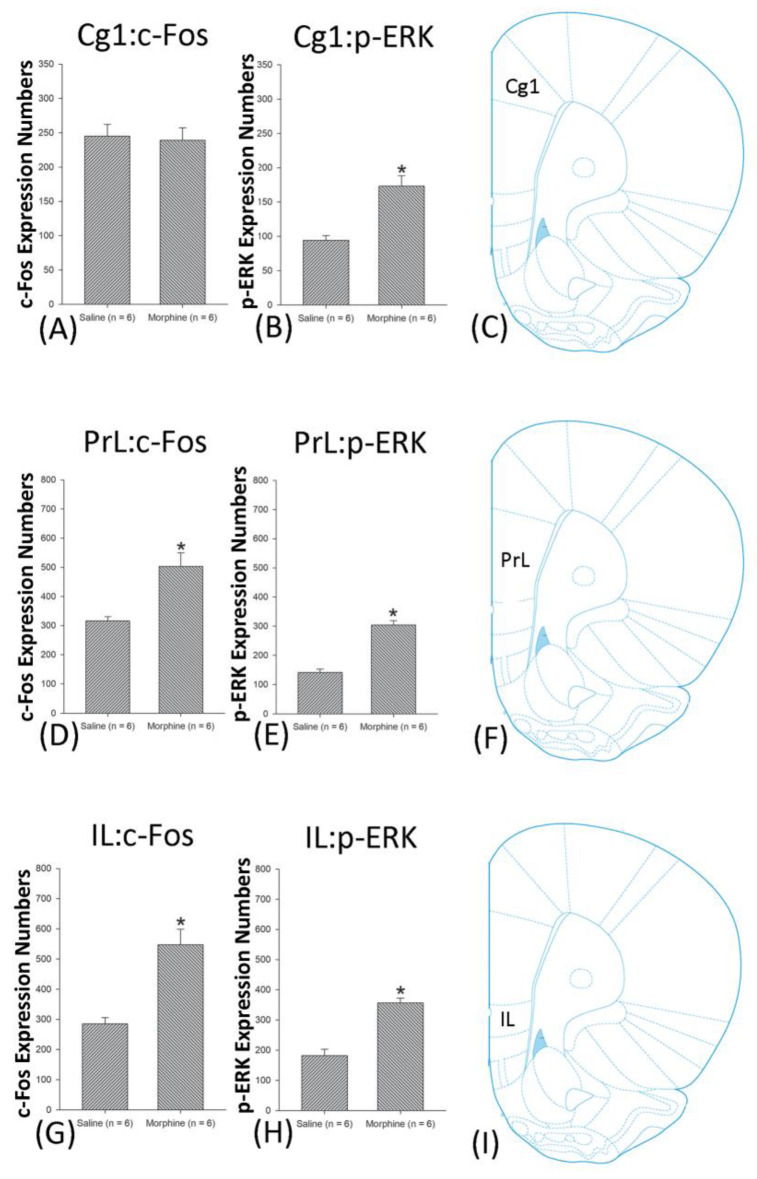
c-Fos or p-ERK expression in the cingulated cortex area 1 (Cg1), the prelimbic cortex (PrL), and the infralimbic cortex (IL) in reinstatement. (**A**) c-Fos and (**B**) p-ERK expression in the Cg1, (**D**) c-Fos and (**E**) p-ERK expression in the PrL, and (**G**) c-Fos and (**H**) p-ERK expression in the IL for reinstatement. (**C**,**F**,**I**) indicate a schematic representation of the Cg1, PrL, and IL. * *p* < 0.05 when comparing the saline and morphine groups (*n* = 6 per group).

**Figure 15 jcm-10-03197-f015:**
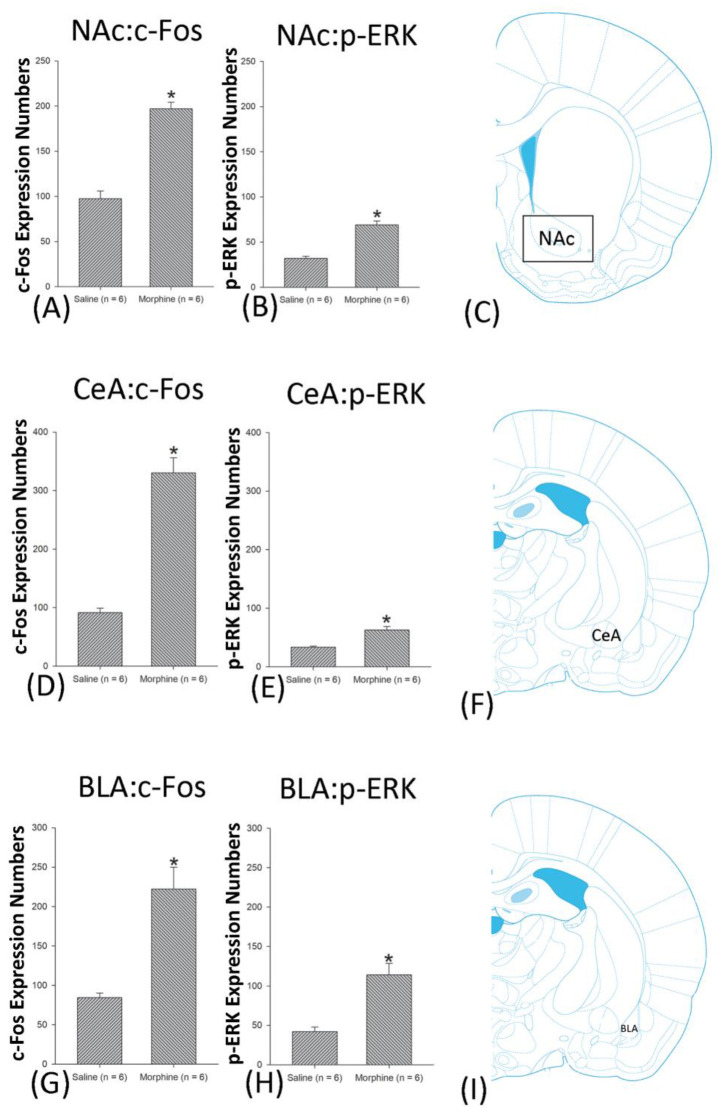
c-Fos or p-ERK expression occurred in the nucleus accumbens (NAc), central amygdala (CeA), and basolateral amygdala (BLA) in reinstatement. c-Fos expression, p-ERK expression, respectively, in the NAc (**A**,**B**), in LH (**D**,**E**), and in PC (**G**,**H**). (**C**,**F**,**I**) indicate a schematic representation of the NAc, CeA, and BLA. * *p* < 0.05 when comparing the saline and morphine groups (*n* = 6 per group).

**Figure 16 jcm-10-03197-f016:**
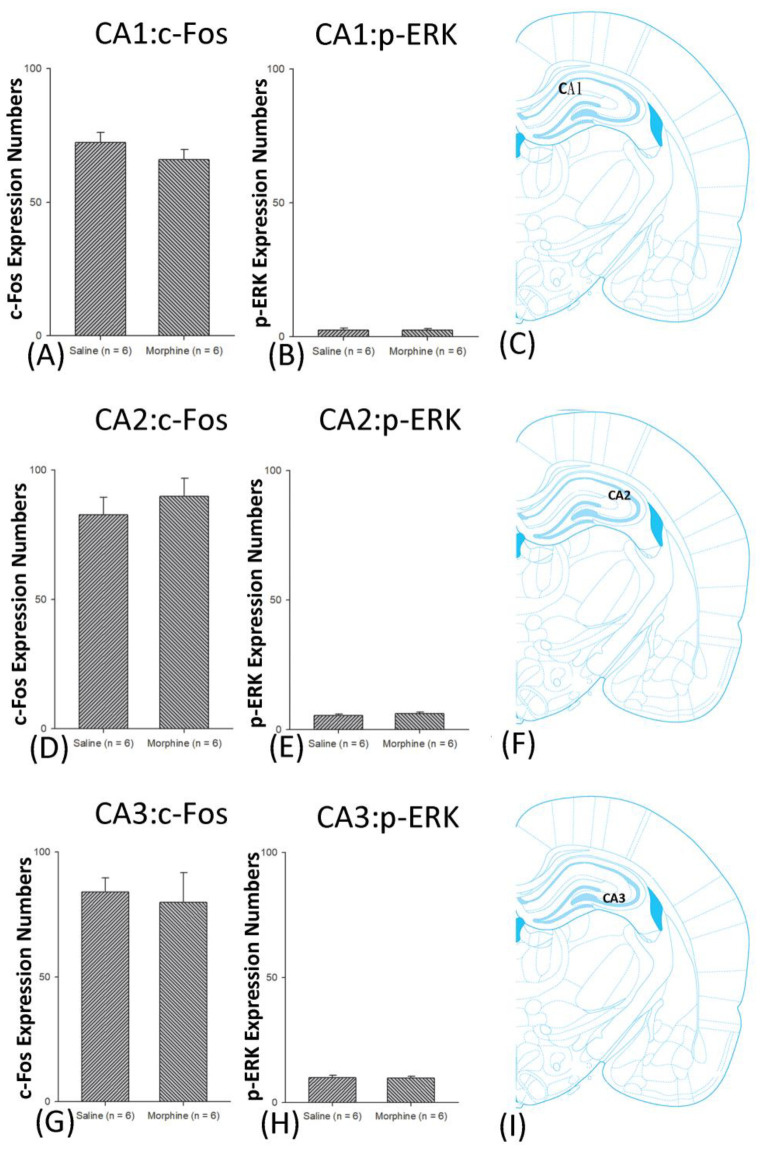
c-Fos or p-ERK immunoreactivity in the CA1, CA2, and CA3 in conditioning. (**A**) c-Fos expression and (**B**) p-ERK expression in the CA1, (**D**) c-Fos and (**E**) p-ERK expression in the CA2, and (**G**) c-Fos and (**H**) p-ERK expression in the CA3 for reinstatement. (**C**,**F**,**I**) indicate a schematic representation of the CA1, CA2, and CA3. * *p* < 0.05 when comparing the saline and morphine groups (*n* = 6 per group).

**Figure 17 jcm-10-03197-f017:**
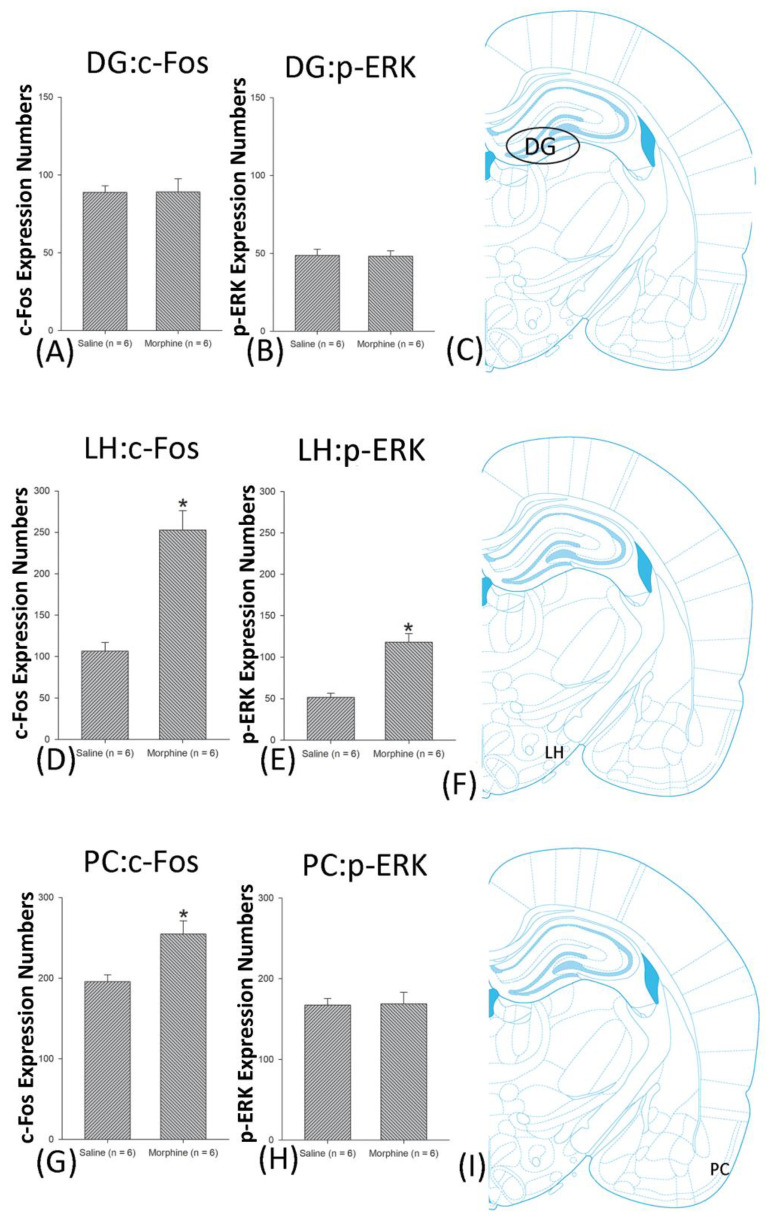
c-Fos or p-ERK expression in the dentate gyrus (DG), lateral hypothalamus (LH), and piriform cortex (PC) in reinstatement. (**A**) c-Fos and (**B**) p-ERK expression in the DG, (**D**) c-Fos and (**E**) p-ERK expression in the LH, and (**G**) c-Fos and (**H**) p-ERK expression in the PC for reinstatement. (**C**,**F**,**I**) indicate a schematic representation of the DG, LH, and PC. * *p* < 0.05 when comparing the saline and morphine groups (*n* = 6 per group).

**Figure 18 jcm-10-03197-f018:**
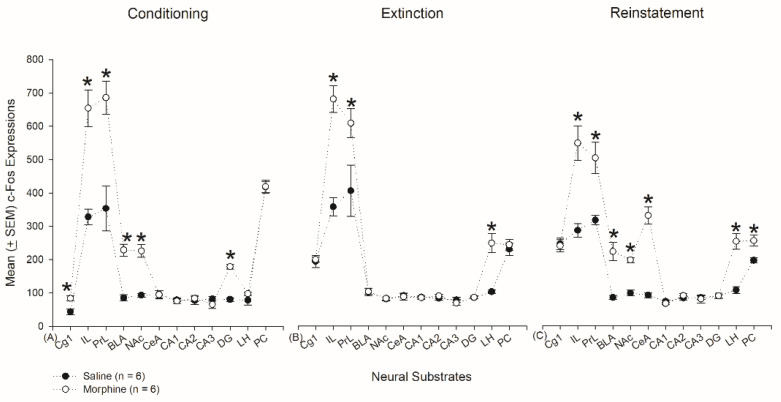
Assessing homeostasis analysis with c-Fos expression for the cingulate cortex area 1 (Cg1), the prelimbic cortex (PrL), the infralimbic cortex (IL), the basolateral amygdala (BLA), the nucleus accumbens (NAc), the central amygdala (CeA), the CA1, the CA2, the CA3, the dentate gyrus (DG), the lateral hypothalamus (LH), and the piriform cortex (PC) in the three phases: (**A**) conditioning, (**B**) extinction, and (**C**) reinstatement. * *p* < 0.05 when comparing the saline and morphine groups (*n* = 6 per group).

**Figure 19 jcm-10-03197-f019:**
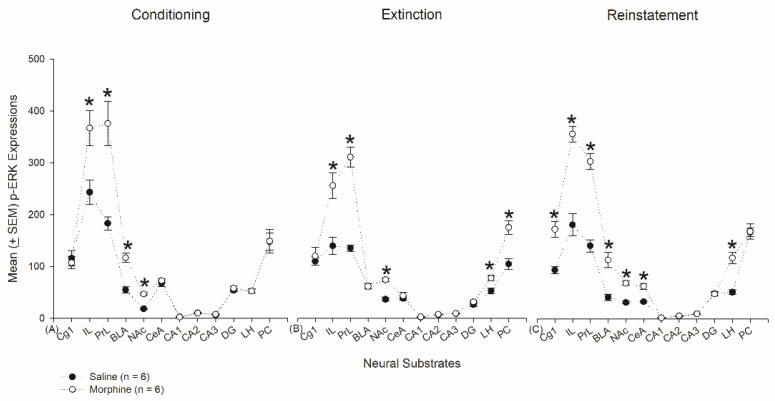
Assessing homeostasis analysis using the p-ERK expression for the cingulate cortex area 1 (Cg1), the prelimbic cortex (PrL), the infralimbic cortex (IL), the basolateral amygdala (BLA), the nucleus accumbens (NAc), the central amygdala (CeA), the CA1, the CA2, the CA3, the dentate gyrus (DG), the lateral hypothalamus (LH), and the piriform cortex (PC) in the three phases: (**A**) conditioning, (**B**) extinction, and (**C**) reinstatement. * *p* < 0.05 when comparing the saline and morphine groups (*n* = 6 per group).

**Figure 20 jcm-10-03197-f020:**
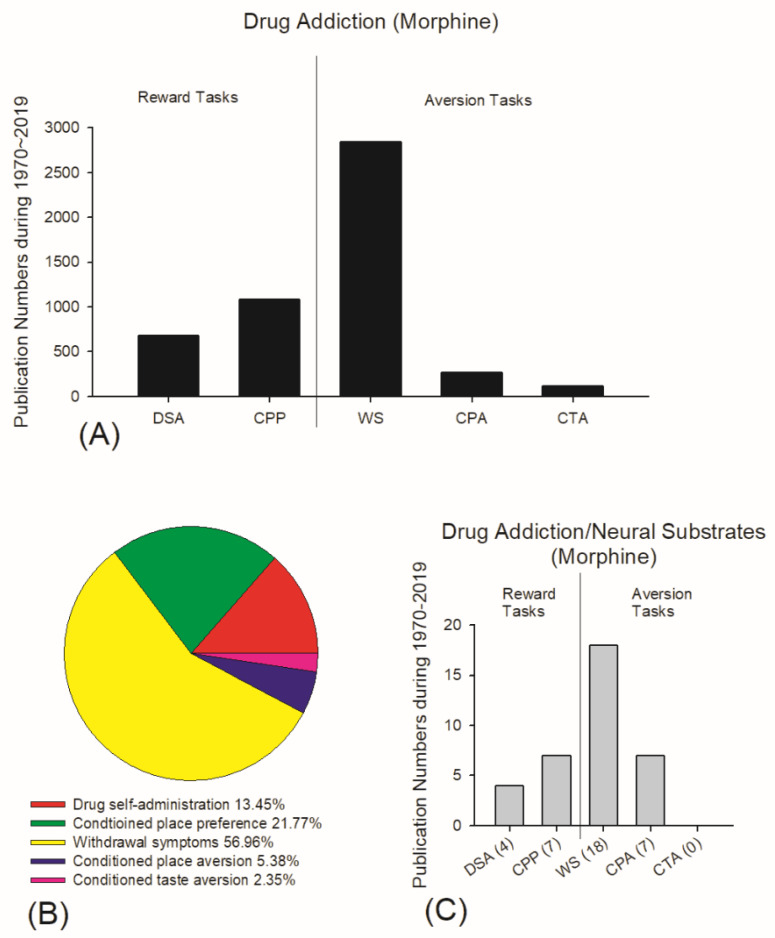
Publications related to morphine addiction in reward and aversive tasks during 1970–2019. (**A**) The number of publications on morphine addiction. Reward tasks include drug self-administration (DSA) and conditioned place preference (CPP). Aversive tasks involve withdrawal symptoms (WS), conditioned place aversion (CPA), and conditioned taste aversion (CTA). (**B**) The percentage of publications on morphine addiction for reward tasks in DSA and CPP and aversive tasks in WS, CPA, and CTA. (**C**) The number of publications on the involvement of neural substrates in morphine addiction for reward tasks in DSA and CPP and aversive tasks in WS, CPA, and CTA.

**Figure 21 jcm-10-03197-f021:**
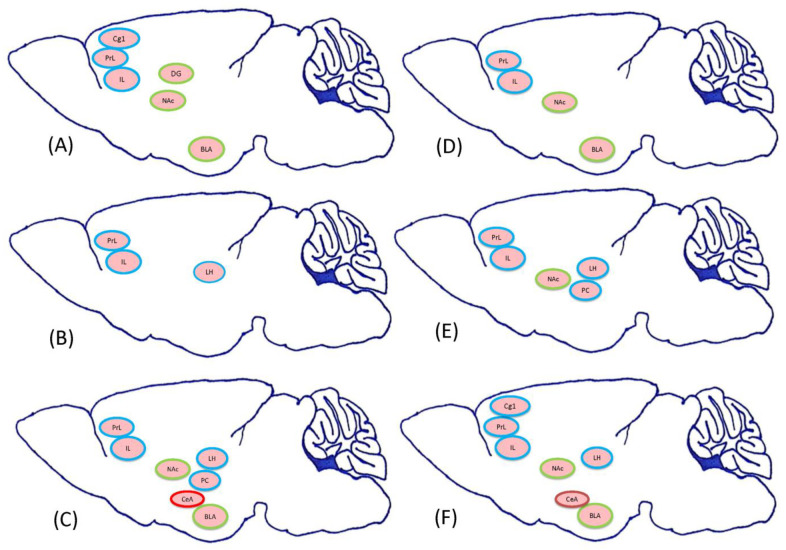
Representation of the neural substrates involved in morphine-induced conditioned suppression of saccharin solution intake for (**A**) conditioning, (**B**) extinction, (**C**) reinstatement to label c-Fos expression or for (**D**) conditioning, (**E**) extinction, and (**F**) reinstatement to label p-ERK. Based on the previous data, the rewarding, aversive, or both properties of neural substrates are, respectively, shown in blue, red, and green in outside circles. The inside filled red ovals represent the neural substrates involved in the aversive CTA effect in the present study. The examined neural substrates include the cingulate cortex area 1 (Cg1), the prelimbic cortex (PrL), the infralimbic cortex (IL), the nucleus accumbens (NAc), the CA1, the CA2, the CA3, the dentate gyrus (DG), the central amygdala (CeA), the basolateral amygdala (BLA), the lateral hypothalamus (LH), and the piriform cortex (PC).

**Table 1 jcm-10-03197-t001:** Summary of various doses of morphine in conditioned taste aversion and conditioned place preference.

	Morphine Doses (mg/kg)				
	0	10	20	30	40
CTA: Aversion	--	+	+	+	+
CPP: Reward	--	--	+	+	+

--, nonsignificant difference; +, significant difference; CTA, conditioned taste aversion; CPP, conditioned place preference.

**Table 2 jcm-10-03197-t002:** Indicating the total of brain regions (12). Numbers and percentages of neural substrates for c-Fos or p-ERK hyperexpression on conditioning, extinction, and reinstatement phases after morphine-induced conditioned suppression of saccharin solution intake.

	Conditioning		Extinction		Reinstatement	
	Numbers	%	Numbers	%	Numbers	%
c-Fos hyperexpression	5	41.67	3	25.00	7	58.33
p-ERK hyperexpression	4	33.33	5	41.67	7	58.33

## Data Availability

https://www.dropbox.com/sh/m9xbk1impl1xlpn/AABunvkcQToDtta0Va2r5cXMa?dl=0 (accessed on 19 July 2021).

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
