# Peer review of "The Paradoxical Effect Hypothesis of Abused Drugs in a Rat Model of Chronic Morphine Administration"

_jcm, 2021, doi:10.3390/jcm10153197_

Round 1
Reviewer 1 Report
GENERAL
I think the idea of showing the paradoxical effects of drugs of abuse, and the fact that you found a dose that induces both preference and aversion, is very cool. I can see you did an enormous amount of work, as well. Therefore, please, sell your paper better: make better figures + shorten your writing.
Major:
You use different volumes in your injections. You need to address this either in the methods or results or somewhere else.
The writing is very long, and repetitive. If you have information in tables and figures, you don’t need to write it, just point it in the text.
You need to structure your Introduction better.
There is missing information in the methods (see specific comments below)
Consider using a Supplementary material file for the details, and keep only the very essential stuff here. Same with the figures. For example, I know it is important and somewhat nice to see the representative pictures, but perhaps this could go in the Suppl. Material and you only keep the barplots/boxplots and the schematic of the brain regions.
Minor:
You call it paradoxical effect hypothesis of abused drugs, but elsewhere I have read it as “dual effect”. So, maybe just add this as a key word.
ABSTRACT (before reading the whole manuscript)
L17: replace “possess” by “induces”
L18: replace “were” by “are”
L20: in which model? Mice? Rats?
L20-21: which second experiment?
L29-30: which concentrations of morphine you are talking about here?
L32: you have two verbs… maybe you are missing “, and” between “CTA” and “resulted”?
INTRODUCTION
You need to work on the structure of your introduction. Usually, you should start by giving general background information (e.g., on drug abuse neuroscience), and then move into your subject (in this case, morphine), and then talk about the hypothesis and experiments.
L54: which first experiment? Is it in the literature or you mean in this study?
L56: what is CPP?
L80: same here, which second experiment? Is this in this study? If so, I would recommend not mentioning this here and giving only the background information here. Then you might mention this experiment info in your last paragraph.
METHODS
L100-127: Nice, I like it is very detailed. However, if you need to shorten the manuscript, don’t hesitate on putting some of this information as a supplementary material.
L28: Finally I know what experiment 1 and experiment 2 are !
L129: Figures 1 and 2 are no flowcharts but results…
L141: delete “each day”. It is redundant.
Experiment 1: it is very well explained, but perhaps you could make a small figure showing a maze picture or at least a schematic figure of the maze, and the time line of the experiment.
L177 & L188: how many rats were used in experiment 2? How many brains were used for c-Fos and p-ERK?
L207: was given, injected?
L221: cite ImageJ program
L222: you mean the number of neurons? You did it for the whole-brain slice or by brain region? --- make clear what you mean by “numbers”… after the whole manuscript, I think you mean number of brain regions.
You are missing information on the microscope you used to visualize the slices.
L229: Mmm there you have a confounding factor: you are injecting different volumes, so in theory you should not be able to compare your control (1ml/kg) with the treatments of 2, 3 and 4 ml/Kg. You have to make this clear here, and make a statement in the discussion why you think the extra saline injected is not causing the observed effect. Next time, inject the same volume.
L355-357: even if you found non significance, you should publish these results as a supplementary material.
RESULTS
Figures 5 to 13: the axes are so small that I cannot read them. I cannot see the scale in the pictures. Also, why using barplots instead of boxplots?
L378: delete “shows”, L485: delete “indicates”, and so on…
Figure 14 & 15: take off the continuous line, x is discrete here.
Table 2: indicate the total of regions (12)
L538-561: what are “Orders”? Where do “Orders” come from? Mention in the methods.
L594-598: if you have this information in a table, you don’t need to write it, just point to the table.
DISCUSSION
L690 & L699: where are the data / results for LiCl-induced CTA?
L727: you should mention in the methods how you did your review.
L775-783: Is this not mentioned in the introduction? Consider moving to the intro.
Figure 17: this would be best in the introduction.
Figure 18 & 19: I think you can enhance these figures a lot! For instance, make the circles bigger and fill them with blue, green, or red because just the outlines are difficult to observe. Also, consider merging both figures to safe space i.e., placing the figures next to each other.
Author Response
Comments and Suggestions for Authors
GENERAL
I think the idea of showing the paradoxical effects of drugs of abuse, and the fact that you found a dose that induces both preference and aversion, is very cool. I can see you did an enormous amount of work, as well. Therefore, please, sell your paper better: make better figures + shorten your writing.
Major:
Point 1: You use different volumes in your injections. You need to address this either in the methods or results or somewhere else.
Response 1: Thank you for your comment. We have revised this point, and we made the description clear in the Methods. Please see 2.6 Drugs of the Method section.
“…2.5. Drugs
Sodium saccharin and sodium chloride were bought from the Sigma-Aldrich Company in the USA. Sodium saccharin was dissolved in distilled water and prepared in a 0.1% saccharin solution. Morphine hydrochloride was bought from the Food and Drug Administration, Ministry of Health and Welfare, Executive Yuan in Taipei in Taiwan. The control group was given 0 mg/kg morphine, and this group was intraperitoneally injected with normal saline with 1 ml/kg. Morphine was dissolved in normal saline at a concentration of 10 mg/ml. Morphine was intraperitoneally injected at a 1, 2, 3, or 4 ml/kg volume, and then they were transferred and served as the doses of 10, 20, 30, and 40 mg/kg morphine. Although the injection volumes of morphine were different, the doses were injected 10, 20, 30, and 40 mg/kg into the body, respectively.…”
Point 2: The writing is very long, and repetitive. If you have information in tables and figures, you don’t need to write it, just point it in the text.
Response 2: Thank you for your comment. We have revised this point. Please see the whole article.
Point 3: You need to structure your Introduction better.
Response 3: Thank you for your comment. We have revised the structure of the Introduction section.
Point 4: There is missing information in the methods (see specific comments below)
Consider using a Supplementary material file for the details, and keep only the very essential stuff here. Same with the figures. For example, I know it is important and somewhat nice to see the representative pictures, but perhaps this could go in the Suppl. Material and you only keep the barplots/boxplots and the schematic of the brain regions.
Response 4: Thank you for your comments. We have followed your comments to revise this point. Please see Supplementary materials.
Minor:
Point 5: You call it paradoxical effect hypothesis of abused drugs, but elsewhere I have read it as “dual effect”. So, maybe just add this as a key word.
Response 5: Thank you for your comment. We have modified the key words and added “dual effect” as a key word. Please see keywords.
“…Keywords: morphine, the paradoxical effect hypothesis of abused drugs, reward, aversion, dual effect…”
ABSTRACT (before reading the whole manuscript)
Point 6: L17: replace “possess” by “induces”
Response 6: Thank you for your comment. We have revised it. Please see L17 in Abstract section.
“…However, whether morphine induces reward and aversion and which neural substrates are involved in morphine’s reward and aversion remains unclear…”
Point 7: L18: replace “were” by “are”
Response 7: Thank you for your comment. We have revised it. Please see L18 in Abstract section.
“…However, whether morphine induces reward and aversion and which neural substrates are involved in morphine’s reward and aversion remains unclear…”
Point 8: L20: in which model? Mice? Rats?
Response 8: Thank you for your comment. We have revised it. Please see L20 in Abstract section. We have added “in rats”.
“…The present study first examined which doses of morphine can simultaneously produce reward in conditioned place preference (CPP) and aversion in conditioned taste aversion (CTA) in rats…”
Point 9: L20-21: which second experiment?
Response 9: Thank you for your comment. We have revised it. Please see L20-21 in Abstract section.
“…Furthermore, the aversive dose of morphine was determined. Moreover, using the aversive dose of 10 mg/kg morphine tested plasma corticosterone (CORT) levels and examined which neural substrates were involved in the aversive morphine-induced CTA on conditioning, extinction, and reinstatement…”
Point 10: L29-30: which concentrations of morphine you are talking about here?
Response 10: Thank you for your comment. We have added 10 mg/kg morphine in L29-30, and also we revised this sentence. Please see L29-30.
“…A dose of 10 mg/kg morphine was only induced the aversive CTA, and it produced higher plasma CORT levels in conditioning and reacquisition but not extinction. High plasma CORT secretions by 10 mg/kg morphine-induced CTA most likely result from stress-related aversion but were not a rewarding property of morphine…”
Point 11: L32: you have two verbs… maybe you are missing “, and” between “CTA” and “resulted”?
Response 11: Thank you for your comment. We have revised this point and added “, and” between “CTA” and “resulted”. Please see L32 in the Abstract section.
“…For assessments of c-Fos and p-ERK expression, the cingulate cortex 1 (Cg1), prelimbic cortex (PrL), infralimbic cortex (IL), basolateral amygdala (BLA), nucleus accumbens (NAc), and dentate gyrus (DG) were involved in the morphine-induced CTA, and resulted from the aversive effect of morphine on conditioning and reinstatement…”
INTRODUCTION
Point 12: You need to work on the structure of your introduction. Usually, you should start by giving general background information (e.g., on drug abuse neuroscience), and then move into your subject (in this case, morphine), and then talk about the hypothesis and experiments.
Response 12: Thank you for your comment. We have revised this point. Please see the whole Introduction section.
Point 13: L54: which first experiment? Is it in the literature or you mean in this study?
Response 13: Thank you for your comment. The study includes two experiments. Please see the descriptions in 2.3 Behavioral procedure of the Method section and L129-135.
Point 14: L56: what is CPP?
Response 14: Thank you for your comment. CPP is conditioned place preference. Because we change the structure of the Introduction section, we have added the full name in the first term of CPP. Please see the first paragraph of the Introduction section.
Point 15: L80: same here, which second experiment? Is this in this study? If so, I would recommend not mentioning this here and giving only the background information here. Then you might mention this experiment info in your last paragraph.
Response 15: Thank you for your comment. We have followed your comments to revise this point. Please see L80. Also, we have mentioned the first and second experiments information in the last paragraph.
“…Furthermore, the previous studies showed that the subareas Cg1 [19], PrL [18], and IL [27] of the mPFC; CA1, CA2, and CA3 of the hippocampus [28], the LH [29], and the PC [30] contributed to the reinforcement process or the rewarding effect of the abused drugs. Moreover, the DG of the hippocampus [26], the NAc [31], and the BLA [32] are involved in the reward and aversion of abused drugs. Growing evidence shows that the CeA only mediates the aversion of abused drugs for drug addiction [33, 34] (Figure 1). In Experiment 2, the aversive dose of morphine tested whether the selected neural substrates (such as the mPFC [e.g., Cg1, PrL, and IL], the hippocampus [e.g., CA1, CA2, CA3, and DG], the NAc, the LH, the amygdala [e.g., BLA and CeA], and the PC) mediate morphine-induced conditioned taste suppression in conditioning, extinction, and reinstatement using c-Fos and p-ERK immunohistochemical staining…”
“…Altogether, the study concerned (a) which doses of morphine can simultaneously produce reward in CPP and aversion in CTA tasks in Experiment 1. (b) Experiment 2 used the aversive dose of morphine to examine which neural substrates were involved in the aversive morphine-induced conditioned suppression and testing of corticosterone levels in conditioning, extinction, and reinstatement. (c) Analyzing c-Fos and p-ERK expression demonstrated the paradoxical effect—reward and aversion and nonhomeostasis or disturbance by morphine-induced conditioned taste suppression…”
METHODS
Point 16: L100-127: Nice, I like it is very detailed. However, if you need to shorten the manuscript, don’t hesitate on putting some of this information as a supplementary material.
Response 16: Thank you for your comment. We have followed to revise this point and put some of this information as a supplementary material. Please see Supplementary Materials.
Point 17: L128: Finally I know what experiment 1 and experiment 2 are !
Response 17: Thank you for your comment.
Point 19: L129: Figures 1 and 2 are no flowcharts but results…
Response 19: Thank you for your comment. We have drawn flowcharts for Experiment 1 in Figure S1 and Experiment 2 in Figure S2. Please see Supplementary Materials.
Point 20: L141: delete “each day”. It is redundant.
Response 20: Thank you for your comment. We have followed your comment to delete “each day”. Please see L141 in “2.2.1.Experiment 1: Testing different doses of morphine for reward and aversion”.
“…During this period, the rats were deprived of water for 23.5 h/day…”
Point 21: Experiment 1: it is very well explained, but perhaps you could make a small figure showing a maze picture or at least a schematic figure of the maze, and the time line of the experiment.
Response 21: Thank you for your comment. We have followed your comments to add the flowchart of Experiment 1. Please see Figure S1.
Point 22: L177 & L188: how many rats were used in experiment 2? How many brains were used for c-Fos and p-ERK?
Response 22: Thank you for your comment. We have written a paragraph to explain how many rats were used in Experiment 2. How many brain were used for c-Fos and p-ERK. Please see “2.2.2.Experiment 2: Which neural substrates were involved in the reward, aversion, or even both effects induced CTA by morphine”. The total number of rats used in Experiment 1 and Experiment 2 has been modified in “Animals of Supplementary Materials and Methods”.
“…Experiment 2 used 132 rats. Note that, 48 rats were tested the behavioral tasks in conditioning (n = 16), extinction (n = 16), and reinstatement (n = 16) phases. Seventy-two rats were conducted by immunohistochemical staining to label c-Fos (n = 36) and p-ERK (n =36) expression in conditioning, extinction, and reinstatement, and each of conditioning, extinction, and reinstatement phases used 12 rats for saline and morphine groups (n = 6 per group), respectively. Twelve rats were conducted using the Enzyme-Linked Immunosorbent Assay (ELISA) corticosterone assessment for the saline (n = 6) and morphine groups (n = 6) in baseline, conditioning, extinction, and reinstatement phases…”
“…For this study, 202 male Wistar rats (weighing 250-300 g at the beginning of the experiment; BioLasco Taiwan Co., Ltd.) were housed in a colony room, with a pair of rats in each plastic home cage (47 cm long × 26 cm wide × 21 cm high) and with hardwood laboratory bedding (Beta Chip)…”
Point 23: L207: was given, injected?
Response 23: Thank you for your comment. We have modified the word. Please see “Immunohistochemical staining: c-Fos and p-ERK” in the Method section.
“…2.4. Immunohistochemical staining: c-Fos and p-ERK
The rats were injected with a sodium pentobarbital overdose. Later, 100 ml of 0.1 M sodium phosphate-buffered saline (PBS) solution was injected…”
Point 24: L221: cite ImageJ program
Response 24: Thank you for your comment. We have cited ImageJ program in the reference. Please see L221. We have cited papers related to InageJ program. Please see “2.4.Immunohistochemical staining: c-Fos and p-ERK”.
“…The ImageJ program counted the positive expression of p-ERK or c-Fos visually at 20 magnifications for each slice for the whole brain [41]…”
Point 25: L222: you mean the number of neurons? You did it for the whole-brain slice or by brain region? --- make clear what you mean by “numbers”… after the whole manuscript, I think you mean number of brain regions.
Response 25: Thank you for your comment. We have corrected this point. Please see “2.4.Immunohistochemical staining: c-Fos and p-ERK”.
“…The number of c-Fos or p-ERK protein expressions was averaged in a selected brain area for each group...”
Point 26: You are missing information on the microscope you used to visualize the slices.
Response 26: Thank you for your comment. We have followed your comment to revise this point. Please see “2.4.Immunohistochemical staining: c-Fos and p-ERK”.
“…The ImageJ program counted the positive expression of p-ERK or c-Fos visually at 20 magnifications for each slice for the whole brain [41]…”
Point 27: L229: Mmm there you have a confounding factor: you are injecting different volumes, so in theory you should not be able to compare your control (1ml/kg) with the treatments of 2, 3 and 4 ml/Kg. You have to make this clear here, and make a statement in the discussion why you think the extra saline injected is not causing the observed effect. Next time, inject the same volume.
Response 27: Thank you for your comment. We have made the description clear and made a statement in “2.5 Drugs” in the Method section.
“…Morphine was dissolved in normal saline at a concentration of 10 mg/ml. Morphine was intraperitoneally injected at a 1, 2, 3, or 4 ml/kg volume, and then they were transferred and served as the doses of 10, 20, 30, and 40 mg/kg morphine. Although the injection volumes of morphine were different, the doses were injected 10, 20, 30, and 40 mg/kg into the body, respectively.”
Point 28: L355-357: even if you found non significance, you should publish these results as a supplementary material.
Response 28: Thank you for your comment. We have followed your comment to draw the representative photomicrographs of c-Fos and p-ERK immunoreactivity for CA1, CA2, and CA3 in the supplementary figures (Figure S5, Figure S9, Figure S13) for the data of nonsignificantly different c-Fos or p-ERK expression. Moreover, we have added to draw the immunohistochemical staining figures of CA1, CA2, and CA3 for c-Fos and p-ERK expression in Figures 8, 12, and 16. Please see Figure S5, S9, and S13 in the Supplementary Materials and Figures 8, 12, and 16.
RESULTS
Point 29: Figures 5 to 13: the axes are so small that I cannot read them. I cannot see the scale in the pictures. Also, why using barplots instead of boxplots?
Response 29: Thank you for your comment. We have added the c-Fos and p-ERK data for nonsignificant differences of brain areas in CA1, CA2, and CA3. So, the Figure’s number has been modified. The axes of the original Figures 5 to 13 have been modified. Please see Figures 6 to 17. On the other hand, we draw the barplots but not boxplots because the data of c-Fos and p-ERK were not big variance. These data were very stable. Thus, the barplots were suitable for the c-Fos and p-ERK data.
Point 30: L378: delete “shows”, L485: delete “indicates”, and so on…
Response 30: Thank you for your comment. We have followed your comments to delete them. Please see all Figure legends.
Point 31: Figure 14 & 15: take off the continuous line, x is discrete here.
Response 31: Thank you for your comment. Although the x is discrete but not continuous line, we would like to see the pattern of a variety of neural substrates in c-Fos- or p-ERK expression for the whole brain. Thus, I have replaced the continuous line with dots-line to show the nonhomeostasis and disturbance among all selected brain areas. Please see Figure 18 and Figure 19.
Point 32: Table 2: indicate the total of regions (12)
Response 32: Thank you for your comment. We have followed your comment to add the description in Table 2. Please see Table 2.
“…Table 2. indicate the total of brain regions (12). Numbers and percentages of neural substrates for c-Fos or p-ERK hyperexpression on conditioning, extinction, and reinstatement phases after morphine-induced conditioned suppression of saccharin solution intake…”
Point 33: L538-561: what are “Orders”? Where do “Orders” come from? Mention in the methods.
Response 33: Thank you for your comment. We have followed your comment to mention the description of “Orders of the trend-line in trend analysis” in the Method section. Please see “2.7 Statistical analysis” of the Method section.
“…one-way repeated trend analysis were conducted in the conditioning, extinction, and reinstatement phases for saline and morphine groups to analyze the trend-line, respectively. Trends are identified by drawing lines, called trend lines. The different orders of trend lines indicate the different shapes of trends, such as one order trend-line for linear, two orders trend-line for quadratic, three orders trend-line for cubic, etc…”
Point 34: L594-598: if you have this information in a table, you don’t need to write it, just point to the table.
Response 34: Thank you for your comment. We have followed your comments to revise this point. Please see L594-598.
“…Table 2 depicts that analysis of the number and percentage of neural substrates for c-Fos or pERK expression during conditioning, extinction, and reinstatement…”
DISCUSSION
Point 35: L690 & L699: where are the data / results for LiCl-induced CTA?
Response 35: Thank you for your comment. We have deleted the description of LiCl-induced CTA. Please see L690 and L699.
Point 36: L727: you should mention in the methods how you did your review.
Response 36: Thank you for your comment. We have added a paragraph “2.8. Review” in Supplementary Materials. Please see “Review” of Supplementary Materials.
“…Review
In discussion, the present study used the Pubmed database to review publications related to morphine addiction for reward and aversion tasks during 1970-2019…”
Point 37: L775-783: Is this not mentioned in the introduction? Consider moving to the intro.
Response 37: Thank you for your comment. We have moved L775-783 to the Introduction section. Please see the Introduction section.
Point 38: Figure 17: this would be best in the introduction.
Response 38: Thank you for your comment. We have followed your comments to revise this point. Figure 17 has been moved to the Introduction section to become Figure 1.
Point 39: Figure 18 & 19: I think you can enhance these figures a lot! For instance, make the circles bigger and fill them with blue, green, or red because just the outlines are difficult to observe. Also, consider merging both figures to safe space i.e., placing the figures next to each other.
Response 39: Thank you for your comment. We have revised this point and merged figures into one figure. Please see Figure 21.

Reviewer 2 Report
This is a very interesting study that needs some correction before publication.
A number of problems with the use of English are present - although they are easily corrected. I will not list all - but here are some:
Title:
"The paradoxical effect hypothesis of abused drugs in morphine" is incomplete
I suggest something like:
"The paradoxical effect hypothesis of abused drugs in a rat model of chronic morphine administration."
Although, you could also refer to a rat model of morphine addiction - if the model is recognized. In any case, as written, the title is a problem.
Abstract:
"To combine with previous drug addiction data, morphine injections may induce hyperactivity in many neural substrates, which mediate reward and/or aversion, due to disturbance and nonhomeostasis in the brain."
Would better read as:
"In the context of previous drug addiction data, the evidence suggests that morphine injections may induce hyperactivity in many neural substrates, which mediate reward and/or aversion, due to disturbance and nonhomeostasis in the brain."
Line 92
(a). w
delete '.'
Line 114
2.2.1.Lickometer.
needs a space after 2.2.1. and Lickometer.
Check other headings below
lines 236 - 238
These sentences do not make sense:
"If need, Tukey’s honestly significant difference post hoc test was carried out for each session. For CPP, the mean spent a dependent t-test analyzed time for drug-unpaired and drug-paired sides among 0-40 mg/kg morphine groups.
Perhaps, "If need be..."
rephrase "the mean spent a dependent t-test analyzed time'
Line 968
"Riley and colleagues reported that abused drugs might have a rewarding effect and 968 aversive [19]. "
Better worded as:
"Riley and colleagues reported that abused drugs might have a rewarding and aversive effects [19]."
There may be other problems with language and typing - which are common in a manuscript of this length. These can often be corrected in proof if someone fluent in English reads over the final draft.
Lastly, in the introduction, it may be worth referring to mu-opioid receptor as belonging to the G protein coupled receptor (GPCR) class of receptors.
This portion is confusing and needs to be restructured:
The last sentence of this paragraph is very important and should be the first sentence of the net paragraph describing the work, thus:
"The first experiment addressed this issue and tested various doses of morphine for aversion in a CTA task and reward in a CPP task. Previous studies of abused drugs have shown that certain brain areas mediated reward or aversion or even both effects. For example..."
This is a long paragraph - but so long as the text is all on topic, this is okay (see my corrections to line 68-71 below)
The next paragraph then introduces the second experiment - making the structure easier to understand and read quickly.
On line 54, at the end of the paragraph is a good place to add:
"Morphine studied in the present work, binds to the mu-opioid receptor expressed on....neurons and belongs to the G protein coupled receptor." (GPCR) class of receptors."
Cite X) Miles D Thompson, Takeshi Sakurai, Innocenzo Rainero, Mary C Maj, Jyrki P Kukkonen. Orexin Receptor Multimerization versus Functional Interactions: Neuropharmacological Implications for Opioid and Cannabinoid Signalling and Pharmacogenetics. Pharmaceuticals (Basel) . 2017 Oct 8;10(4):79. doi: 10.3390/ph10040079.
Other language problems in introduction:
Line 68 -71
Fewer neural substrates are mainly contributed to the aversive effect in drug addiction. In particular to the central amygdala (CeA), the corticotrophin-releasing factor secretions involved anxiety or aversion on withdrawal symptoms [13].
Try this:
"Fewer neural substrates are contribute to the aversive effect in drug addiction. With respect to the central amygdala (CeA), corticotrophin-releasing factor secretions are involved anxiety or aversion on withdrawal symptoms [13]."
Author Response
Comments and Suggestions for Authors
This is a very interesting study that needs some correction before publication.
A number of problems with the use of English are present - although they are easily corrected. I will not list all - but here are some:
Point 1: Title:"The paradoxical effect hypothesis of abused drugs in morphine" is incomplete. I suggest something like: "The paradoxical effect hypothesis of abused drugs in a rat model of chronic morphine administration.". Although, you could also refer to a rat model of morphine addiction - if the model is recognized. In any case, as written, the title is a problem.
Response 1: Thank you for your comment. We have revised this point. Please see the title.
Abstract:
Point 2: "To combine with previous drug addiction data, morphine injections may induce hyperactivity in many neural substrates, which mediate reward and/or aversion, due to disturbance and nonhomeostasis in the brain."
Would better read as:
"In the context of previous drug addiction data, the evidence suggests that morphine injections may induce hyperactivity in many neural substrates, which mediate reward and/or aversion, due to disturbance and nonhomeostasis in the brain."
Response 2: Thank you for your comment. We have revised this point. Please see the Abstract section.
Point 3: Line 92
(a). w
delete '.'
Response 3: Thank you for your comment. We have deleted “.”. Please see Line 92.
Point 4: Line 114
2.2.1.Lickometer.
needs a space after 2.2.1. and Lickometer.
Response 4: Thank you for your comment. We have moved this paragraph into the Supplementary Materials. Please see the Supplementary Materials.
Check other headings below
Point 5: lines 236 - 238
These sentences do not make sense:
"If need, Tukey’s honestly significant difference post hoc test was carried out for each session. For CPP, the mean spent a dependent t-test analyzed time for drug-unpaired and drug-paired sides among 0-40 mg/kg morphine groups.
Perhaps, "If need be..."
rephrase "the mean spent a dependent t-test analyzed time'
Response 5: Thank you for your comment. We have revised them. Please see the Statistical analysis of the Supplementary Materials.
Point 6: Line 968
"Riley and colleagues reported that abused drugs might have a rewarding effect and 968 aversive [19]. "
Better worded as:
"Riley and colleagues reported that abused drugs might have a rewarding and aversive effects [19]."
Response 6: Thank you for your comment. We have followed you comment to revise it. Please see Lin 968.
Point 7: There may be other problems with language and typing - which are common in a manuscript of this length. These can often be corrected in proof if someone fluent in English reads over the final draft.
Response 7: Thank you for your comment. We have revised it. Please see the whole article.
Point 8: Lastly, in the introduction, it may be worth referring to mu-opioid receptor as belonging to the G protein coupled receptor (GPCR) class of receptors.
This portion is confusing and needs to be restructured:
The last sentence of this paragraph is very important and should be the first sentence of the net paragraph describing the work, thus:
"The first experiment addressed this issue and tested various doses of morphine for aversion in a CTA task and reward in a CPP task. Previous studies of abused drugs have shown that certain brain areas mediated reward or aversion or even both effects. For example..."
This is a long paragraph - but so long as the text is all on topic, this is okay (see my corrections to line 68-71 below)
Response 8: Thank you for your comment. We have modified a little bit. Please see The Introduction section.
Point 9: The next paragraph then introduces the second experiment - making the structure easier to understand and read quickly.
Response 9: Thank you for your comment. We have revised it. Please see the Introduction section.
Point 10: On line 54, at the end of the paragraph is a good place to add:
"Morphine studied in the present work, binds to the mu-opioid receptor expressed on....neurons and belongs to the G protein coupled receptor." (GPCR) class of receptors."
Cite X) Miles D Thompson, Takeshi Sakurai, Innocenzo Rainero, Mary C Maj, Jyrki P Kukkonen. Orexin Receptor Multimerization versus Functional Interactions: Neuropharmacological Implications for Opioid and Cannabinoid Signalling and Pharmacogenetics. Pharmaceuticals (Basel). 2017 Oct 8;10(4):79. doi: 10.3390/ph10040079.
Response 10: Thank you for your comment. We found another better place to add your suggestion. Please see the Introduction section.
“…Morphine studied in the present work, binds to the mu-opioid receptor expressed on opiate neurons and interacts with the orexin receptor belongs to the G protein-coupled receptor, activating orexin signaling [15]…”
Point 11: Other language problems in introduction:
Line 68 -71
Fewer neural substrates are mainly contributed to the aversive effect in drug addiction. In particular to the central amygdala (CeA), the corticotrophin-releasing factor secretions involved anxiety or aversion on withdrawal symptoms [13].
Try this:
"Fewer neural substrates are contribute to the aversive effect in drug addiction. With respect to the central amygdala (CeA), corticotrophin-releasing factor secretions are involved anxiety or aversion on withdrawal symptoms [13]."
Response 11: Thank you for your comment. We have followed your comment to revise it. Please see the Introduction section.

Reviewer 3 Report
This is an interesting study conducted by Yu et al. on the paradoxical effect hypothesis of abused drugs in morphine. I only have few comments to make.
1) The role of traditionally known pathways of morphine such as PAG and morphine receptors are not mentioned in the manuscript. They should be mentioned. Also, in line 1011 authors talked that opioid receptors could mediate the different actions of morphine. Please provide additional explanation.
2) In the study, the higher doses of morphine induced both reward in CPP and aversion in CTA. What may be the reason? It is not sufficiently explained in the manuscript.
3) In addition, the relation of the results obtained from the study and its future use in clinics should be discussed. In what way the treatment should apply the results?
4) It would be better for the readers to present the behavioral time table/schedule in figures.
Author Response
Comments and Suggestions for Authors
This is an interesting study conducted by Yu et al. on the paradoxical effect hypothesis of abused drugs in morphine. I only have few comments to make.
Point 1:The role of traditionally known pathways of morphine such as PAG and morphine receptors are not mentioned in the manuscript. They should be mentioned. Also, in line 1011 authors talked that opioid receptors could mediate the different actions of morphine. Please provide additional explanation.
Response 1: Thank you for your comment. We have followed your comment to revise this point. Please see “4.10. Further studies” in the Discussion section.
“…Previous studies have demonstrated that the periaqueductal gray matter (PAG) was also involved in conditioned aversive learning [61, 62] and aversive behaviors induced by morphine or opiate drugs. For example, the PAG lesion interfered with morphine-induced CTA [61]. Microinjections of the mu-opioid receptor agonist morphine or the kappa-opioid receptor agonist U-50488H in the dorsal PAG caused the CPA effect, indicating the mu- or kappa-opioid receptors mediated the CPA learning [62]. In an elevated plus-maze test, morphine microinjections (like benzodiazepine compound, midazolam) in the dorsal PAG increased entries and spent time in the open arm, and morphine-induced anti-anxiety/anti-aversive effect was reversed by a systematic injection of mu-opioid antagonist naloxone [63]. Therefore, the PAG was seemingly to mediate morphine-induced aversive conditioning and aversive behaviors…”
Point 2: In the study, the higher doses of morphine induced both reward in CPP and aversion in CTA. What may be the reason? It is not sufficiently explained in the manuscript.
Response 2: Thank you for your comment. We have written the possible mechanism and explanation in the Discussion section. Please see “4.10. Further studies” in the Discussion section.
“…This evidence might explain why the higher doses of morphine could trigger the rewarding effect in CPP and the aversive effect in CTA. The high dose of morphine may simultaneously activate the mu-opioid receptor pathway to produce the rewarding effect in CPP and trigger the kappa-opioid receptor mechanism to induce the aversive effect in CTA. This issue of morphine’s aversive and rewarding mechanisms should be examined in further studies.…”
Point 3: In addition, the relation of the results obtained from the study and its future use in clinics should be discussed. In what way the treatment should apply the results?
Response 3: Thank you for your comment. We have followed your comment to revise this point. Please see “4.11. Clinical implication of the Discussion section”.
“…4.11. Clinical implication
Investigations of brain mechanisms related to the paradoxical effect of abused drugs could provide new insights or contribute to new interventions for use in drug-addiction clinics. The present findings may indicate that hyperactivity of the brain is distributed in many places, resulting in disturbance and nonhomeostasis and involving various rewarding and aversive neural substrates. Drug abusers experience euphoria or happiness from the rewarding property of abused drugs and suffer the stressful or aversive effect. Drug addiction seems to be a complicated process and involves paradoxical effects of different neural substrates in reward and aversion. However, these paradoxical effects of reward and aversion cannot cancel each other within the brain. The abuser might experience distress and mental disturbance during morphine treatments. Based on present data, recovering the homeostasis of the brain is essential, as is developing novel pharmacological and nonpharmacological interventions to ameliorate the symptoms of drug addiction effectively. Insight from these findings applies in the clinical treatment of drug addiction…”
Point 4: It would be better for the readers to present the behavioral time table/schedule in figures.
Response 4: Thank you for your comment. We have revised this point. Please see Figures S1 and S2.

This manuscript is a resubmission of an earlier submission. The following is a list of the peer review reports and author responses from that submission.